# weather@home 2: validation of an improved global-regional climate modelling system

Benoit P. Guillod[1], Richard G. Jones[1,3], Andy Bowery[2], Karsten Haustein[1], Neil R. Massey[1], Daniel M. Mitchell[1], Friederike E. L. Otto[1], Sarah N. Sparrow[2], Peter Uhe[1,2], David C. H. Wallom[2], Simon Wilson[3], and Myles R. Allen[1]

[1]Environmental Change Institute, University of Oxford, Oxford, United Kingdom
[2]Oxford e-Research Centre, University of Oxford, Oxford, United Kingdom
[3]Met Office Hadley Centre, Exeter, United Kingdom

*Correspondence to:* Benoit P. Guillod (benoit.guillod@ouce.ox.ac.uk)

**Abstract.** Extreme weather events can have large impacts on society and, in many regions, are expected to change in frequency and intensity with climate change. Owing to the relatively short observational record, climate models are useful tools as they allow for generation of a larger sample of extreme events, to attribute recent events to anthropogenic climate change, and to project changes of such events into the future. The modelling system known as weather@home, consisting of a global climate model (GCM) with a nested regional climate model (RCM) and driven by sea surface temperatures, allows to generate very large ensemble with the help of volunteer distributed computing. This is a key tool to understanding many aspects of extreme events. Here, a new version of weather@home system (weather@home 2) with a higher resolution RCM over Europe is documented and a broad validation of the climate is performed. The new model includes a more recent land-surface scheme in both GCM and RCM, where subgrid scale land surface heterogeneity is newly represented using tiles, and an increase in RCM resolution from 50 km to 25 km. The GCM performs similarly to the previous version, with some improvements in the representation of mean climate. The European RCM temperature biases are overall reduced, in particular the warm bias over eastern Europe, but large biases remain. Precipitation is improved over the Alps in summer, with mixed changes in other regions and seasons. The model is shown to represent main classes of regional extreme events reasonably well and shows a good sensitivity to its drivers. In particular, given the improvements in this version of the weather@home system, it is likely that more reliable statements can be made with regards to impact statements, especially at more localised scales.

## 1   Introduction

Anthropogenic climate change due to increased greenhouse gases concentration in the atmosphere poses numerous threats to society (IPCC, 2013). In particular, the frequency, intensity, and duration of extreme events such as heat waves, droughts and flooding may have already changed due to climate change (Frich et al., 2002; Fischer and Knutti, 2015), a trend that is expected to continue in the future (Seneviratne et al., 2012). The growing field of extreme event attribution attempts to answer the question whether and to what extent anthropogenic climate change altered the frequency and intensity of observed extreme events. Answering this question is now becoming possible for many events (National Academies of Sciences, Engineering,

and Medicine, 2016), and is done by quantifying the role of anthropogenic climate change versus natural climate variability for events that have occurred in the past (e.g., Otto et al., 2012; Stott et al., 2016). Another field of research investigates how extreme events may change in the future, thereby concentrating on future climate projections (e.g., Mitchell et al., 2016c).

Owing to their rarity, extreme weather events and their characteristics can be difficult to assess. Indeed, only a few such events may be available in observational records. Therefore, model-based approaches consisting of large ensembles that allow for the statistics of rare events to be analysed are an essential complement to observational products. In particular, large ensembles of Global Climate Models (GCMs) allow derivation of multiple sequences of weather patterns and a substantial number of associated extreme events. Dynamical downscaling of these GCM simulations by Regional Climate Models (RCMs, Giorgi, 2006) can provide more spatially detailed information, which can be very valuable for the investigation of localised impacts of extreme weather events.

One such modelling system is weather@home (Massey et al., 2015). Consisting of a GCM with prescribed sea surface temperatures (SSTs) and sea ice and a nested RCM over a region of interest, it leverages the computing power of volunteers around the world to generate very large ensembles of GCM-RCM simulations. This is particularly useful for the investigation of extreme weather events, and weather@home has been used successfully for the attribution of many extreme weather events (e.g., Pall et al., 2011; Otto et al., 2012) as well as their impacts, such as flooding-related property damages (Schaller et al., 2016) and heat-related mortality (Mitchell et al., 2016b).

Model performance is however a common limitation inherent to modelling approaches. Like any model, weather@home exhibits biases in certain variables (Massey et al., 2015). In particular, a substantial warm and dry bias was found in summer over Eastern Europe, similar to many RCMs (Jacob et al., 2007). The increase in capabilities of home computers, on which weather@home simulations are being run, makes it possible to increase the model resolution and include newer model developments, with the aim to reduce these biases.

Although identifying the causes of GCM and RCM biases is not straightforward, previous studies suggest that the land surface may play an important role (e.g., Davin et al., 2016), in particular for summer climate. Several studies have identified the tendency for RCMs to display an excessive summer drying over Europe (Christensen et al., 2007; Kotlarski et al., 2014). The resulting dry summer soil moisture bias in turn feeds back onto the atmosphere through the underestimation of evapotranspiration (or latent heat flux) and the simultaneous overestimation of the sensible heat flux at the surface (e.g., Seneviratne et al., 2010). These fluxes may directly affect temperature (via the sensible heat flux) and precipitation (via moisture input to the atmosphere, e.g., Eltahir and Bras, 1996). In addition, they can lead to indirect effects modulated by the boundary layer, thereby affecting cloud cover (e.g., Ek and Holtslag, 2004) and precipitation (e.g., Findell and Eltahir, 2003; Taylor et al., 2011; Guillod et al., 2015). Although weather@home biases could be due to atmospherically-driven lack of precipitation, improvements to land-surface schemes in RCMs and GCMs have been shown to substantially improve the simulated surface climate (e.g., Davin et al., 2011, 2016), suggesting that at least part of the biases may be attributable to deficiencies in the representation of the land surface. Besides model formulation, other aspects of the land surface such as soil parameters may significantly impact surface climate (e.g., Guillod et al., 2013).

A new version of weather@home (called weather@home 2) was therefore developed by including a more recent version of the land-surface model MOSES (see Sect. 2.2). In this paper, we describe and validate the GCM globally and the RCM over the European domain, with a focus on the simulation of mean climate, daily extremes and the reliability of the model response to forcings.

The paper is structured as follows: In Sect. 2, we describe weather@home and the new developments that lead to its second version, as well as the modelling simulations and observational data used in this paper. The GCM (HadAM3P) is validated in Sect. 3, with a focus on mean biases in temperature, precipitation and atmospheric circulation. Section 4 provides a detailed validation of the RCM (HadRM3P) over Europe, including analyses of the model biases in mean and extremes as well as its reliability. Section 5 draws some conclusions on the suitability of the modelling system to investigate extreme weather events.

## 2   Model description and experiments

### 2.1   weather@home

The climate modelling system known as weather@home (Massey et al., 2015) is part of the climate modelling project *climateprediction.net* (Allen, 1999). It consists of an Atmospheric GCM, HadAM3P, that is downscaled to a higher resolution over a limited domain by its RCM equivalent, HadRM3P. The downscaling is only coupled one-way, so that the RCM can not impact on the GCM. Both models share essentially the same physics and are based on the atmospheric component of the coupled climate model of the UK Met Office Hadley Centre, HadCM3 (Gordon et al., 2000), with a number of improvements described in Massey et al. (2015). These include increasing the GCM horizontal resolution to $1.875° \times 1.25°$ (in longitude and latitude, respectively) and introducing better representations of large-scale and convective clouds. The formulation of the RCM, HadRM3P, differs from HadAM3P only in terms of horizontal resolution, time step (reduced from 15 to 5 minutes) and resolution-dependent physical parameters. In general, HadRM3P is run on a rotated grid allowing it to simulate the area of interest over an equatorial domain (in the rotated coordinate system) at quasi-uniform horizontal resolutions of $0.44°$ or $0.22°$. It has been run over many regions world-wide, including all of those defined by CORDEX (Coordinated Regional Climate Downscaling Experiment) initiative (Giorgi et al., 2009), although any domain can be specified. HadRM3P is run alongside a given HadAM3P simulation, the latter providing the lateral boundary conditions at the regional domain edges at 6 hourly intervals.

Both models are forced with sea surface temperature (SST) and sea ice, atmospheric composition, $SO_2$ emissions (including volcanoes), and solar forcing, as well as initial conditions for all model variables. The GCM HadAM3P has been shown to represent the atmospheric dynamics well compared with many state-of-the-art GCMs (Mitchell et al., 2016a).

The strength of weather@home resides in its ability to run very large ensembles of simulations, of the order of thousands to tens of thousands. To achieve this, volunteer distributed computing via the Berkeley Open Infrastructure for Network Computing (BOINC, Anderson, 2004) is used. Individual simulations are sent to volunteers around the world, who run the HadAM3P-HadRM3P simulations and upload the results on a server. A large number of simulations can thereby be performed in parallel which are particularly relevant when examining extreme events, rare by definition and requiring large numbers of simulated

years to define their statistics robustly. The weather@home project has led to many high-impact analyses, notably in the field of extreme event attribution, where sets of simulations with observed or with corresponding "natural" conditions (without anthropogenic climate change) can be compared to assess the role of human influences on extreme events (e.g., Schaller et al., 2016; Mitchell et al., 2016b; Haustein et al., 2016).

While the weather@home project initially focused on a European region (e.g., Massey et al., 2015) and North American region (the Pacific North-west, Li et al., 2015; Mote et al., 2016) it has also been successfully used in Australia and New Zealand (Black et al., 2016), Africa (Marthews et al., 2015), and is currently also being deployed over a number of additional regions.

## 2.2    Model developments for version 2 of weather@home

A few modifications have been incorporated in version 2 of weather@home (hereafter w@h2) relative to the original weather@home (hereafter w@h1, described in detail by Massey et al., 2015). More specifically, a more recent land-surface scheme was introduced in both HadAM3P and HadRM3P, and the standard horizontal resolution of HadRM3P was increased.

In both model versions, HadRM3P is run over the European CORDEX domain (Fig. 1). Currently, HadRM3P in w@h1 has always been run at a horizontal resolution of $0.44°$ (about 50 km, see Fig. 1b) over Europe, while in w@h2 resolution has
been increased to $0.22°$ (about 25 km, see Fig. 1a). As mentioned in Sect. 2.1, any domain and resolution can in principle be specified – the resolutions mentioned here refer to the standard configurations used in weather@home as well as in the simulations analysed in this study (see Sect. 2.3).

The main development included in w@h2 is an improved representation of the land surface. In w@h2, the land-surface model (LSM) MOSES 1 used in w@h1 (Cox et al., 1999) was replaced by a more sophisticated version, MOSES 2 (Essery
et al., 2003). MOSES is a third-generation LSM, incorporating the direct physiological effect of $CO_2$ on photosynthesis and stomatal conductance (Sellers et al., 1997). The total land evapotranspiration includes interception evaporation from the canopy, plant transpiration, bare soil evaporation, and snow sublimation. Five vegetation types and four non-vegetated surface types are considered. The soil is represented by 4 layers spanning a total depth of 3 m, with the hydrology following Richards' equation (see Cox et al., 1999, for further details).

The main difference between the two LSM versions is the explicit consideration of land surface heterogeneities within each grid cell via the introduction of a tiling scheme in MOSES 2 (Essery et al., 2003). Indeed, in MOSES 1 only one surface type is considered in each grid cell. The introduction of tiles in MOSES 2 allows consideration of each of the nine surface types mentioned above, and computation of surface fluxes for each surface type, of which the area-weighted average is returned to the atmospheric component of the model.

Another improved representation of the land-surface introduced into w@h2 is the dynamic vegetation model TRIFFID (Top-Down Representation of Interactive Foliage and Flora Including Dynamics, Cox, 2001). The vegetation distribution (i.e., fraction of surface types within each grid cell) in MOSES 2 can be either prescribed to observed values or computed interactively by TRIFFID. In w@h2, TRIFFID has been implemented in the regional but not in the global model. Although for

most applications TRIFFID is switched off and both models are similar in that respect, a side effect is that the prognostic snow albedo cannot be turned on in the global model, while it is turned on by default in the regional model.

In addition to the tiling scheme, a number of smaller improvements have been implemented in MOSES 2, notably in the representation of snow processes (Essery and Clark, 2003).

Finally, the definition of the region over which the RCM is run is more flexible in w@h2 than in w@h1. While in w@h1 one application was built and deployed for each region separately, w@h2 consists of a single executable that can be used for any region, the latter being defined via input parameters. This simplifies the extension of weather@home to many regions, although the creation of an initial condition file remains necessary for any newly created region.

## 2.3 Modelling experiments

A large ensemble of w@h2 consisting of more than 100 simulations per year from 1900–2006 is analysed. First, a restart file from a century-long HadAM3P simulation with MOSES 1 has been reconfigured for MOSES 2. This initial condition file is then used in a spin-up ensemble consisting of 12-month simulations (from December to November, with multiple simulations for each year), providing spun-up initial conditions on December 1st each year. The simulations analysed in this paper are then initialised on the 1st of December each year from the end state of the spin-up ensemble and are run for 13 months. The effect of
the relatively short spin-up for soil variables on simulated temperature and precipitation is discussed in Sect. 3.1 for HadAM3P and Sect. 4.1 for HadRM3P. The correspondence between simulated and real years comes from using observed sea-surface temperature and sea-ice as the lower boundary condition and observed concentrations of greenhouse gases, SO2 emissions and influence of volcanoes and solar radiation.

The sea surface temperature and sea ice are prescribed from observed estimates in the HadISST dataset (Rayner et al., 2003)
version 2.1.0.0 (see Titchner and Rayner, 2014, for sea ice), a pre-release version directly provided by the UK Met Office Hadley Centre. The other input variables of greenhouse gases concentrations ($CO_2$, $CH_4$, $N_2O$, $O_3$ and halocarbon gases), $SO_2$ emissions, volcanic activity and solar forcing are prescribed to historical values as in Massey et al. (2015) with the data also provided by the Met Office Hadley Centre.

To assess whether the model developments described in Sect. 2.2 lead to an improved representation of climate in w@h2
compared to w@h1, we also use the w@h1 ensemble from Massey et al. (2015), consisting of about 20 members per year from 1961–1990. It should be noted that the difference between these two model ensembles may not only result from the models themselves, but also from ($i$) differences in the prescribed SSTs and sea ice, the analysed w@h1 ensemble being based on version 1 of the HadISST dataset (Rayner et al., 2003), and ($ii$) horizontal resolution, this latter point applying only to the RCM.

## 30 2.4 Observational data

We use gridded observation-based climate products for the model validation. Global temperature and precipitation over land (excluding Antarctica) are taken from version 3.23 of the Climate Research Unit time series dataset (CRU–TS; Harris et al., 2014), covering 1901–2014, which we interpolate to the model grid using a first-order conservative scheme. Global atmospheric

fields (geopotential height) are taken from the Japanese 55-year Reanalysis (JRA-55) project carried out by the Japan Meteo-rological Agency (Kobayashi et al., 2015) and are bilinearly interpolated to the model grid. For the validation of HadRM3P, we use the E–OBS dataset (Haylock et al., 2008) version 12.0, which provides daily temperature and precipitation data on the model grid from 1950 to present. To validate the land surface fluxes in HadRM3P, we use two datasets available over the

common time period 1984–2006: The satellite-based dataset Surface Radiation Balance (SRB) version 3.1 (Stackhouse et al., 2004; Zhang et al., 2015) is used for surface radiation fluxes, and the FLUXNET-MTE product (Jung et al., 2009, 2011) is used for surface sensible and latent heat fluxes. These two datasets are bilinearly interpolated to the rotated RCM grid.

## 3   Global model validation

In this Section, we investigate the performance of HadAM3P in w@h2. First, seasonal mean biases in surface air temperature,

precipitation and geopotential height at 500hPa (as a proxy for the background state of atmospheric flow) are shown and compared to those in w@h1 over a 30 years period from 1961–1990 (Sect. 3.1; w@h2 biases look very similar when the whole time period, from 1900–2006, is considered) and are complemented by biases in variability. Then, time series of global land temperature and precipitation are shown and discussed in Sect. 3.2.

### 3.1   Seasonal mean biases

Figure 2 shows the ensemble mean seasonal biases in surface air temperature in w@h2 (left; a,d,g,j) and in w@h1 (centre; b,e,h,k) relative to CRU-TS, as well as the difference between the absolute bias values (right; c,f,i,l; these are expressed so that negative values, in green, indicate an improvement in w@h2 compared to w@h1). Overall, the bias patterns are similar in both model versions, with the largest biases found in the Northern Hemisphere winter (December to February, DJF) and summer (June to August, JJA). The difference between the biases in the two models is most prominent in JJA, with significant

improvements over Africa, the southern US, and parts of central Russia. Conversely, the biases in that season are higher in w@h2 in the north of North America, eastern Russia, and western Russia and Europe. The improved land surface scheme in HadAM3P therefore does not improve the representation of summer temperature averages over Europe (Fig. 2i). In DJF, the difference between the two models is smaller, with w@h2 performing slightly better than w@h1 in the whole southern hemisphere but slightly poorer over eastern North America, north Africa and India. In the northern hemisphere spring (March

to May, MAM), biases are larger in w@h2 over the eastern US, Canada and parts of Asia, but reduced over Europe, western and north Russia, Alaska and India. The difference between the two models is small in September to November (SON), with improvements in the Southern Hemisphere and mixed differences in the Northern Hemisphere. Table 1 summarises the biases globally, expressed as area-weighted root mean squared biases. Globally, the performance is very similar in both models, with a small improvement for all seasons in w@h2 compared to w@h1. For most regions, the performance of HadAM3P is similar

to state-of-the-art coupled climate models from CMIP5 (Flato et al., 2013), although a fair comparison is difficult given that in w@h2 the ocean state is prescribed to observations while in CMIP5 models it is computed interactively by an ocean model coupled to the atmospheric model.

Since variability is very relevant for attribution (Uhe et al., 2016), we also compute biases in the standard deviation of monthly averaged temperature (Supplementary Fig. S1). While biases in temperature variability are similar in both model versions, w@h2 tends to improve the representation of summer and autumn monthly variability at mid-latitudes.

The precipitation biases, shown in Fig. 3, highlight some improvements in w@h2 relative to w@h1. In particular, biases are reduced in the rainy season over the Amazon (DJF and MAM) and Africa. These improvements are confirmed by Table 1, with constant or improved biases at the global scale in all seasons. Nonetheless, these improvements are rather small in amplitude and the main biases in w@h1 are still present in w@h2 (Fig. 3a-k). Quite striking are the large dry biases over and around Indonesia in all seasons. Since absolute precipitation biases are dominated by regions with large amounts of rainfall, we also show these biases in relative terms in Supplementary Fig. S2. Apart from the dry areas, which by definition tend to show large relative changes, Fig. S1 highlights the summer dry bias over Eurasia. Differences between w@h1 and w@h2 (Fig. S1c,f,i,l) highlight substantial improvement in w@h2 over East Asia in DJF, as well as over Northern Africa in most seasons. Like with temperature, the model performs similarly to typical CMIP5 models (Flato et al., 2013, but note that as for temperature, this comparison may not be fair given the prescribed SSTs in w@h2 as opposed to interactive ocean in CMIP5). Biases in variability (Supplementary Fig. S3) exhibit similar patterns as biases in mean.

Critical for many extreme events is the state of the atmospheric circulation, features of which are known to be poorly reproduced in current generation climate models (Anstey et al., 2013; Harvey et al., 2014). For instance, strong anticyclonic air advecting from low-latitudes can cause persistent, stable systems over western Europe during summer, leading to extremely hot and dry conditions (e.g., Pfahl and Wernli, 2012). Here, we use seasonal-mean geopotential height at 500 hPa as a proxy for the background atmospheric wave activity (Fig. 4). For a more detailed analysis of the dynamics in w@h1 see Mitchell et al. (2016a, b).

Figure 4 shows that the largest anomalies in the Northern Hemisphere with respect to the reanalysis are during winter. The bias patterns are similar in both models w@h1 and w@h2. This is unsurprising, because capturing mid-latitude jet variability is linked with model resolution (Berckmans et al., 2013), and while the regional model of w@h2 has increased horizontal resolution compared with w@h1, there is no two-way feedback with the global model, so any increase in model resolution will not improve the global atmospheric dynamics. Consequently, no improvement in capturing geopotential height is seen in the Northern Hemisphere. The only major difference between the two model versions are seen in the Southern Hemisphere, in particular over the JJA and SON seasons. However, this is most likely not due to the model version but rather to the use of different SST datasets. Indeed, HadISST2 (used in w@h2) exhibits lower SSTs in the Southern Hemisphere compared to HadISST1 used in w@h1 (not shown). Winter geopotential height variability underestimation as well as summer variability over Europe are improved in w@h2 (Supplementary Fig. S4), but the improvements are overall small – likely also due to the use of the same GCM resolution in both models.

Finally, to assess whether the 1-year spin-up was sufficient to allow the soil variables to be spun-up, Supplementary Fig. S5 shows the difference between ensemble mean soil moisture (for each soil layer) in December between the 1st month and the 13th month of the analysed simulations (i.e., 13th and 25th month of simulation, respectively), scaled by the standard deviation of the second one. Apart from North Africa, the differences are confined to the 3rd (Central Asia) and 4th layer (many regions).

This suggests that a longer spin-up may required in future experiments with w@h2. Fortunately, however, the upper 1m of the soil, corresponding to the root zone in most regions and therefore most critical for evapotranspiration, appears relatively well spun-up over Europe. Nonetheless, the soil moisture state in deeper layers may in some cases impact soil moisture dynamics in the root zone and, thereby, affect land-atmosphere exchange and surface climate. It is not possible to further assess whether an additional year would lead to further changes, as these are not available, and soil temperature is not examined here as this variable has not been saved in our simulations. The impact on temperature biases is shown in Supplementary Fig. S6 and the largest impact is found in DJF but is unlikely due to soil moisture as it spans all latitudes. The most striking difference is a reduction of the bias over Southeast Europe and Central US, which may be driven by increased soil moisture in these regions with soil moisture-limited evapotranspiration regimes (Seneviratne et al., 2010) and possibly by effects of soil temperature. An impact is also found in MAM. This suggests that a longer spin-up might potentially further reduce the summer temperature warm model bias. For precipitation (Supplementary Fig. S7), the impact is small globally, in all seasons except DJF and, in other seasons, over Sahara (note that % biases are very sensitive to small changes in this region). DJF impacts are found throughout latitudes and are thus unlikely to be a soil moisture spin-up issue but may result from changes in circulation induced by temperature changes. These results highlight that a longer spin-up may be required in future uses of w@h2, which will be implemented for future w@h2 experiments.

## 3.2   Global land time series

Given the use of the model for attribution, another interesting question is whether the model is able to simulate the response to external forcings, such as $CO_2$, aerosols and volcanoes. In this section, we focus on the global mean response over land and show time series of global land yearly averages in temperature and precipitation (Fig. 5 for anomalies relative to 1961–1990; see Fig. S8 for raw values). The interquartile (25–75%) and 5–95% ranges of the w@h2 ensemble members for each year provide an estimate of the unpredictable (chaotic) component of atmospheric variability, while variations between years depict the response to the model forcings (including SSTs and sea ice). For temperature, years with strong positive or negative anomalies often match between the observations and the model, and CRU-TS mostly lies within the 90% confidence interval of the w@h2 ensemble (71% of the years, suggesting that variability at the global scale might be slightly underestimated). The global trend also seems well captured, such as the faster warming since the 1980s. Although this may not be surprising since others have found that prescribing SSTs may strongly force trends over land (e.g., Shin and Sardeshmukh, 2011), we note that regional trends computed from various ensemble members suggest a large range of trends despite the prescription of SSTs (see Sect. 4.4). The actual temperature values (Fig. S8a) are very similar to the anomalies (Fig. 5a). For precipitation (Fig. 5b), some discrepancy is found between about 1915–1945, when the model simulates too much rainfall, but observational error is also likely larger in this period. Although CRU-TS appears to lie more often outside the w@h2 ensemble for precipitation than for temperature (observed values are within the 5–95% range from w@h2 on only 58% of the years), some of the spikes (e.g., mid 1950s, early 1970s, late 1990s) and troughs (e.g., mid 1960s, early 1990s) are found in both model and observations, suggesting that HadAM3P is able to reproduce some of the sensitivity of precipitation to drivers such as SSTs. It should be

noted, however, that unlike for temperature, the long term precipitation average is substantially lower in the model than in observations (Fig. S8b), indicating larger biases at the global scale.

Similar time series plots for the 26 SREX regions (Seneviratne et al., 2012) are shown in Figs. S9–S12. Overall, variability from year to year is well captured by the model, suggesting a good model sensitivity to SSTs, greenhouse gases and other drivers. Some regions show a strong dependence of temperature and precipitation on the underlying SST patterns, especially over the tropics (most regions in South America, Africa, and South and Southeast Asia), as opposed to other regions where most of the model spread appears to be due to internal variability within the atmosphere only. These time series suggest that the model's response to external factors is reliable in most regions of the globe.

## 4 Regional model validation

We now move to the validation of the regional climate model HadRM3P within w@h2. As for the validation of HadAM3P in the previous section, we analyse seasonal mean biases in surface air temperature and precipitation and compare these to those in w@h1 over a 30 year period from 1961–1990 (Sect. 4.1). These biases are analysed in detail for the sub-regions shown in Fig. 1, with a focus on the mean biases for regional averages and the geographical distribution of temperature and precipitation within each sub-region. The origin of the mean biases is also investigated in Sect. 4.2. We then look at the ability of the model to represent extremes by means of quantile-quantile plots in Sect. 4.3. The sensitivity of the model to forcings for sub regions within the European domain is then investigated using reliability diagrams (Sect. 4.4).

### 4.1 Mean biases

HadRM3P mean biases in temperature (Fig. 6, with respect to the E-OBS dataset) are similar to those of HadAM3P, including the warm bias in summer. This particular bias, however, is substantially reduced in w@h2 relative to w@h1, over most of central and southeastern Europe in HadRM3P (by 1–2°C, Fig. 6i). This contrasts with results from the GCM HadAM3P for which this bias worsens in this region and season (Sect. 3.1 and Fig. 2i). We note that in w@h1 the summer temperature bias was larger in HadRM3P than in HadAM3P (Fig. 10 in Massey et al., 2015), while in w@h2 the biases are more consistent between the global and regional model. Hence, the improvement in HadRM3P in w@h2 compared to w@h1 comes from not increasing the global model bias.

This improvement could be a result of the higher horizontal resolution in w@h2 (0.22°, versus 0.44° in w@h1), which could explain why this bias is reduced in HadRM3P but not in HadAM3P. The improved representation of the land surface with the introduction of MOSES 2 may also contribute to this improvement, consistently with other studies (e.g., Davin et al., 2016). Feedbacks between the land surface and the atmosphere have indeed been shown to be key to summer temperature in these regions, in particular for hot extremes (e.g., Quesada et al., 2012). The origin of the biases is investigated into greater detail in Sect. 4.2. Probably as a side effect of this bias reduction, the warm bias extends further North in w@h2, inducing a slight degradation of model performance over Scandinavia and Western Russia. Other changes with the introduction of w@h2

include the vanishing of a small warm bias over Central and Eastern Europe in SON, but the appearance of a new small warm bias over Eastern Europe (Ukraine, Bielorussia) in DJF and MAM.

Table 2 shows the biases in regional averages for the 8 regions from the PRUDENCE project (Christensen and Christensen, 2007) shown in Fig. 1. As a complement, Fig. 7 summarises the temperature biases at the grid cell level for the sub-regions expressed as the spatial root mean squared biases (RMSB) in each region. Given that the two regional models are run at different resolutions and that the E-OBS dataset is available on both model grids, RMSB is computed at both resolutions for each model in order to allow for a fair comparison, by bilinearly interpolating w@h1 data to $0.22°$ and aggregating w@h2 data to $0.44°$. The improvement in JJA is found at both resolutions in all regions except Scandinavia (SC), while in other seasons the differences between the two models are found to be rather small at the scale of the analysed regions.

We now examine the biases in precipitation. Figure 8 shows the seasonal mean biases in both model versions and their difference (see Fig. S13 for relative precipitation biases). The biases are very similar between both models. In particular, the dry bias over Eastern Europe in JJA is not reduced in w@h2, which sheds some light on the mechanisms leading to the reduced temperature bias in this region and season. The introduction of the more sophisticated land surface scheme MOSES 2 may impact climate in two main ways: First, MOSES 2 may better simulate evapotranspiration (e.g., by better distributing water across storage components or improved stomatal resistance parameterisation), thereby leading to an improved partitioning of the energy available at the land surface into sensible and latent heat fluxes. Improved surface fluxes, in particular sensible heat flux, directly lead to an improved simulated temperature. Second, altered surface fluxes may additionally impact precipitation (e.g., Gentine et al., 2013; Guillod et al., 2014, 2015), feeding back on the biases. For instance, precipitation may increase as a response to increased evapotranspiration, which may further reduce the biases by providing more water for further evapo-transpiration, thereby leading to cooler and wetter conditions. The absence of an improvement in simulated precipitation over Eastern Europe suggests that this second pathway does not dominate the response. Instead, it is either the direct improvement in simulated evapotranspiration in MOSES 2 or other factors unrelated to the land surface scheme, such as increased horizontal resolution, which reduces temperature biases.

Figure 9 provides an overview of the precipitation biases at the grid cell scale within each sub-regions by showing the precipitation RMSB (as in Fig. 7 for temperature), complemented by Table 2 for the bias of regionally averaged precipitation. Unlike for temperature, model performance for precipitation is highly dependent on horizontal resolution and the interpretation is less straightforward. The region with the largest precipitation biases at the grid cell scale is the Alps (AL). There, the biases are largest for each model at their own resolution, but smaller when interpolated or aggregated to the other resolution. This is expected for w@h2, as aggregating the data to a coarser grid allows for biases of opposing signs in neighbouring grid cells to compensate each other. As a result, w@h2 clearly outperforms w@h1 over the Alps at $0.44°$ resolution. However, the improvement of w@h1 performance after bilinear interpolation to the higher resolution may seem surprising. It suggests that the locations of the peaks in precipitation are shifted relative to the observations, leading to large local biases of both signs within the region, a feature that can indeed be observed in Fig. 8. The geographical distribution of precipitation, quantified by the spatial correlation between seasonally averaged precipitation in model and observations (Fig. S14), highlights that, in most cases, the spatial correlation increases with interpolation or aggregation, while no significant difference between the models

is found at each model's respective resolution. The better resolution of topography thereby does not particularly improve the simulation of spatial patterns within the regions, even over the Alps. The smoothing of the field that results from bilinearly interpolating to $0.22°$ thereby artificially reduces the overall bias. This result is consistent with earlier findings showing that the model exhibits some exaggerated rain-shadow effect (Buonomo et al., 2007), also seen here with a dry bias South of the Alps. This effect also likely plays a role in the better performance of w@h2 at $0.44°$, which should therefore be treated with caution (see, e.g., the apparent improvement in w@h2 found in Fig. 9, where the bias difference is shown at $0.44°$). Nonetheless, it should be noted that for example in JJA, the precipitation bias is halved when considering regional averages over the Alps (Table 2), while no such difference is found at the grid cell scale (Fig. 9), highlighting again the scale-dependency of the biases. This improvement found in JJA at the regional scale, however, does not hold in other seasons. Overall, these results suggest that the analysis of regionally aggregated data in a region may be more appropriate in regions with complex topography than analysis at the grid cell scale.

Finally, the impact of the short spin-up is evaluated as was done in Sect. 3.1 for HadAM3P. Fig. S15 shows the difference in soil moisture as in Fig. S5 (see Sect. 3.1). Over Europe, only Finland and Northwestern Russia display large differences in the upper 1m of the soil. In the deepest layer, soil moisture is larger in the analyzed year than in the previous year over Southeastern Europe and in some other regions, but this deep layer may be less critical to evapotranspiration and therefore to surface climate. Analysis of temperature and precipitation biases (Figs. S16 and S17) show that the hot MAM and JJA biases over Southeastern Europe are reduced with progressing spin-up, as expected from the increasing soil moisture and suggesting that a longer spin-up may further reduce this bias. Temperature biases in DJF and precipitation biases in all seasons are not related to soil moisture changes in a straightforward manner, and hence could be due to soil temperature, a variable not saved as an output in our simulations and therefore not analysed here.

## 4.2 Origin of the biases

To investigate the causes of the biases, and in particular the role of the land surface for these, we analyse surface radiative and turbulent fluxes. Figure 10 shows the seasonal cycle of HadRM3P biases for each region and a number of variables. This analysis was conducted over years 1984–2006 instead of the 1961–1990 period analysed in previous sections due to availability of observations of land surface variables such as radiation (SRB dataset) and surface turbulent fluxes (FLUXNET-MTE dataset). As a side effect, only w@h2 simulations are analysed (the w@h1 ensemble only spans 1961–1990).

The warm and dry summer biases appear clearly on Fig. 10(a,b), in particular over Eastern Europe (EA) and the Mediterranean (MD) regions. Positive biases in net shortwave radiation at the surface (Fig. 10d) are found in most regions from April/May to September, and are mostly driven by an underestimation of cloud cover (not shown; see Massey et al., 2015, for cloud cover biases in w@h1-HadAM3P). The overestimation of incoming energy is most pronounced in June and July in EA, and may explain part of the warm biases.

The turbulent heat fluxes provide further insights into the RCM biases: sensible heat flux ($H$, Fig. 10e), latent heat flux ($\lambda E$, Fig. 10f), and the partitioning of the energy available at the land surface into these two fluxes as expressed by the Evaporative Fraction ($EF = \frac{\lambda E}{\lambda E + H}$), i.e., the fraction of the turbulent fluxes that is used for evapotranspiration. EF (Fig. 10g)

is overestimated in spring but underestimated in summer, a decrease (relative to observations) that is a sign of excessive summer soil moisture depletion. In fact, the overestimation of $\lambda E$ in spring may itself contribute to excessive soil moisture depletion, although precipitation minus evaporation (Fig. 10c) does not exhibit particularly negative biases (note, however, that observed precipitation might be underestimated since the E-OBS dataset does not correct for the systematic undercatch of rain gauge measurements). The result of this drying observed as a bias in EF is ($i$) an overestimation of $H$, particularly in July and August, and ($ii$) a concurrent underestimation of $\lambda E$. The overestimation of $H$ likely contributes to the positive temperature bias in these months. In fact, the MD region appears to be strongly affected by the biases in turbulent fluxes, which may explain its large warm bias despite a radiation bias smaller than other regions such as EA. The underestimation of $\lambda E$, on the other hand, implies a too dry boundary layer, which in turn may lead to an underestimation of cloud cover and precipitation.

These results show that despite the improvements found in w@h2 following the use a more sophisticated land surface scheme, some deficiencies remain. Part of the biases in temperature and precipitation can be explained by the land surface. The origin of these land surface biases could lie in atmospheric parameterisations (e.g., of cloud and precipitation formation), which provide too little precipitation and too high incoming shortwave radiation. Alternatively, deficiencies in the land surface could be the driver of the fast drying of the soils, which in turn feed back onto the atmosphere, leading to the observed cloud, radiation and precipitation biases. A combination of both the atmosphere and the land-surface likely leads to the observed biases, but identifying the driver of these biases is outside the scope of this paper.

## 4.3 Extreme events

The ability of the weather@home ensemble modelling system to generate a large number of simulations makes is particularly attractive for the study of extreme weather events and their attribution to anthropogenic climate change. Various extremes events have been investigated using weather@home, such as floods (Schaller et al., 2016) and heat waves (Otto et al., 2012; Mitchell et al., 2016b). In this section, we analyse the performance of the model for the following extreme events: hot summer days, cold winter nights, and heavy precipitation days in both seasons.

Figures 11 and 12 show quantile-quantile plots for the 8 regions for different variables and seasons, using all overlapping years between E-OBS and our w@h2 ensemble (1950–2006). The dots and crosses contain the values at specific quantiles for the whole ensemble, with filled dots for deciles, empty dots for the values at percentiles 1 to 5 and 95 to 99, and crosses for the 0.5 and 99.5 percentile values. The envelopes provide indications about the spread from ensemble members to assess both uncertainty and internal variability of the model as follows: 1000 bootstraps samples are constructed, each with one ensemble member per year, thereby containing the same total number of days as the observations. The envelope displays the 95% range of the quantile values computed from each bootstrap sample.

We first investigate the performance of hot summer extremes, quantified by the daily maximum temperature (in red in Fig. 11). High daily maximum temperature values are overestimated in all regions. Interestingly, in most regions, the quantiles match the observations very well in the colder half of the data, but not in the warmer tail, highlighting that the warm biases on hot extremes in these regions are responsible for the warm bias in mean temperature. In MD and EA, however, even the cold tail of daily maximum temperature is overestimated. Interestingly, these two regions can be expected to be in a regime where

soil moisture is a major limiting factor to evapotranspiration, thereby strongly controlling summer temperature (e.g., Mueller and Seneviratne, 2012). The dry summer precipitation bias in these two regions (e.g., Fig. 8) can thus be expected to indeed induce a warming over a wide range of temperature quantiles. A possible reason for the bias to be restricted to warm extremes in the other regions may be that the model on some occasions produces a too strong summer drying in these regions, inducing

a shift into a soil moisture-controlled regime and thereby an amplification of temperature anomalies on hot days. Note that the spread from the bootstrap samples is small in most regions, highlighting that these biases do not result from internal variability but are exhibited in any subsample of the same size as observational data.

For cold winter temperatures (daily minimum temperature in DJF, in blue in Fig. 11), the model performs rather well. Apart from the regions MD, SC and, to a lesser extend, ME and AL, where nighttime temperature are underestimated or

overestimated, observed cold quantile values are mostly within the range of the modelled values. Extreme cold night in BI and FR, however, are also underestimated by the model (i.e., extreme cold night are not cold enough). Overall, w@h2 appears to be suitable for the investigation of cold winter nights over Europe.

For daily precipitation (Fig. 12 with JJA in red and DJF in blue), the spread between bootstrap samples is larger. In summer, heavy precipitation days are very well represented in all regions apart from BI and EA, where the quantile values are underes-

timated by w@h2. These regions also exhibit relatively large negative mean precipitation biases (e.g., Table 2). Nonetheless, it appears that overall w@h2 does a reasonable job at simulating summer heavy precipitation extremes in most European regions. Daily winter heavy precipitation (in blue in Fig. 12), on the other hand, is overestimated in most regions (especially in MD, SC, AL, EA), but well simulated in BI, IP, with intermediate performances in FR and ME. We note that unlike for temperature, most precipitation quantile-quantile plots display a rather linear shape, suggesting that for applications where bias correction

is necessary, applying a linear method may be appropriate.

These results provide some confidence in the ability of w@h2 to simulate extreme events over Europe. A few exceptions include summer hot extremes, which are overestimated over all regions. A range of bias-correction methodologies are available to take such biases into account, ranging from a simple additive ("delta method", for temperature) or multiplicative ("linear scaling", for precipitation, e.g., Lafon et al., 2013) adjustment based on the mean, to sophisticated methods that attempt to

correct for changes in the shape of the distribution, such as quantile-quantile mapping (e.g., Wood et al., 2004). The shapes of the quantile-quantile plots for summer daily maximum temperature (Fig. 11) suggest that the application of a simple additive bias correction may not be suitable to correct extremes. A multiplicative factor applied to precipitation, on the other hand, seems appropriate in most regions. However, these bias correction techniques may not preserve the physical consistency between variables that is provided by the model, which may be an issue in the case of impact studies. In the case of large ensembles

such as those from weather@home, a new bias correction methodology (Sippel et al., 2016b), based on the resampling of ensemble members conditional on the distribution of, e.g. summer averaged temperature over a region of interest, has been shown to not only improve seasonal averages, but also the representation of extremes. This new methodology is promising for a wide range of application with weather@home model output.

## 4.4 Reliability and trends

A common use of climate models, including weather@home, is the study of the response of climate to forcing agents. In particular, weather@home is regularly used for the attribution of extreme weather events to anthropogenic climate change. An obvious question is then: Is the model reliable, i.e., does it simulate well the response to potential drivers such as sea surface temperature and greenhouse gases? In this section, we investigate the reliability of w@h2 for simulating seasonally averaged events: warm summers, cold winters, dry summers and wet winters. While seasonal averages are not directly related to extreme weather events, the drivers of both are likely similar (e.g., higher $CO_2$ leads to increased mean and extreme temperature), and the occurrence of a few extreme events may strongly impact the seasonal average. Figures 13 to 16 show reliability diagrams (Weisheimer and Palmer, 2014) for these four types of seasonal events and the 8 analysed regions, using w@h2 and CRU-TS data from 1901–2006. For each type of event (e.g., high summer temperature, defined as JJA averaged temperature in the upper tercile), the probability of the event is computed for each year from regionally averaged w@h2 model output ("forecast probability"). The 106 forecasts (one per year) are then grouped into bins of size 0.1, and the corresponding observed frequency ("observed relative frequency") is computed from the observations in the corresponding years, with uncertainties derived from bootstrapping (Wilks, 2011; NCAR – Research Applications Laboratory, 2015). The forecast and observed values for each bin are then plotted with the size of the dot proportional to the sample size (i.e., number of years). Results for bins containing at least 5 data points (i.e., years) are shown in red, while for other bins, shown in black, values are not very robust and should be interpreted with caution. The grey background in each plot shows the skill region, i.e., where data contribute positively to the Brier Skill Score. Here, we follow a commonly used method (e.g., Weisheimer and Palmer, 2014) whereby the tercile definition is based on the observed and modelled distributions, respectively, i.e., a model's forecast of a warm summer is when the temperature is in the upper tercile (i.e., upper third) of its own distribution.

In order to facilitate interpretation, reliability is further classified in five categories using the definition proposed by Weisheimer and Palmer (2014). To do so, 1000 bootstrap samples with replacement were constructed from the full set of w@h2 data. A reliability diagram was simulated for each of them, to whose points a weighted linear regression was applied, using the number of forecasts in each bin as weights. The 75% confidence interval (uncertainty range) of the regression slopes is used to categorize forecasts into five classes, from 1 (dangerously useless) to 5 (perfect forecast) (Weisheimer and Palmer, 2014). Table 3 provides some detail on the definition of the five categories, and the category is indicated on the upper left of each panel on Figs. 13–16.

Reliability diagrams for warm summers (JJA temperature in the upper tercile, Fig. 13) show that the model is very reliable at simulating the dependency of this quantity to forcings and displays good resolution, albeit with a small underconfidence, i.e., the model tends to overforecast low probability events but underforecast high probability events (see Wilks, 2011, for details on the interpretation of reliability diagrams). Such forecasts can typically perform very well after calibration. Interestingly, this underestimation of the sensitivity of hot temperatures to forcings is consistent with the tendency of RCM to underestimate trends in heat waves over Europe (Min et al., 2013; Sippel et al., 2016a). All regions display skill that is "still very useful for decision making" or "perfect" (categories 4 and 5, respectively). Note that for a few bins (e.g., for forecast values above 0.7 in

IP), observations are in the upper tercile for all years with such forecast (modelled) probabilities, preventing the bootstrapping method to compute uncertainty ranges for individual bins (note that the uncertainty of the linear fit used to categorize the performance can still be applied). In most cases, we also find that data points that lie far from the 1:1 line (e.g., for forecast probabilities greater than 0.4 in FR) include very few years and should therefore be interpreted cautiously (black dots, including

less than 5 years or "forecasts"). A similarly good performance is found for the occurrence of low winter temperature (DJF temperature in the lower tercile, Fig. 14). Thus, the overall high reliability of w@h2 for simulating warm summers and cold winters provides some confidence in weather@home-based attribution statements for temperature over Europe.

The reliability of the model for seasonal averages of precipitation is found to be lower. For low summer precipitation (Fig. 15), the reliability is found to be marginally useful for IP and EA, and not useful for FR. The reliability in other regions is

even lower ("dangerously useless"), as the slope of the linear fit is slightly negative. A more positive picture is found for high winter precipitation: perfect forecasts are identified for ME and SC (Fig. 16), and still marginally useful performance for IP and BI. The reliability is classified as "dangerously useless" for MD, FR and EA (Weisheimer and Palmer, 2014). The relatively low skill for precipitation should however be expected and it is consistent with low seasonal predictability in Europe found in other studies (e.g., Weisheimer and Palmer, 2014).

It should be noted that, as Figs. 13 to 16 are based on 1901–2006, they include the influence of all temporally-varying factors including greenhouse gases, sea surface temperature and sea ice, aerosols and volcanoes. Therefore, these results are dominated by the long-term trend arising from increased greenhouse gas concentrations, rather than by year to year sea surface temperatures variability, for example. Trends in regional averages of temperature and precipitation, quantified using the Theil-Sen slope with Mann-Kendall significance testing (e.g., Yue et al., 2002) are shown in Fig. 17 for summer and winter.

For w@h2, we constructed 1000 106-year time series by randomly sampling one simulation per year, from which trends and p-values are derived. Boxplots summarize these 1000 trend values and are overlaid by white dots depicting the observed trend from CRU-TS. The value at the bottom of each boxplot indicates the percentage of w@h2 time series with significant trend, with an asterisk if the observed trend is significant. Overall, temperature trends are well within the interquartile range of modelled trends, although they are underestimated in IP, FR and AL. Thus, w@h2 follows the tendency of RCMs to underestimates

temperature trends over Europe (Min et al., 2013). For precipitation, on the other hand, trends are noisy and clustered around 0, and often observed trends often lie at the tail of the w@h2 trend distributions. This could explain the overall poor reliability in seasonal averages of precipitation found in Figs. 15 and 16.

Attempts to isolate the response to the oceans (SSTs and sea ice) by using anomalies from a 31 yr running average (not shown) does not provide more insights, as the forecasts from individual years are all close to the climatological forecast of

1/3. This result is consistent with the time series shown on panels (e,k,o) in Figs. S9–S12, which show that for European regions the inter-member spread is substantially larger than the variability in the ensemble mean from year to year (long-term trend excepted). Therefore, in w@h2 most of the inter-annual variability in Europe is due to (unpredictable) internal variability in the atmosphere, rather than to specific SST or sea ice patterns, consistently with the relatively low seasonal predictability often found over Europe (e.g., Weisheimer and Palmer, 2014). Further work will investigate this more specifically and will

aim at determining whether this finding is a model feature or can be confirmed by observations. Here, we simply note that

Figs. S9–S12 suggest a different behaviour in some regions known to be strongly influenced by SSTs patterns such as the El Niño Southern Oscillation.

## 5   Conclusions

The new version of weather@home presented and validated in this paper is a powerful tool for the study of extreme weather events. The modelling setup consists of the GCM HadAM3P driven by sea surface temperature, sea ice and other forcings, which is downscaled over a sub-region by its RCM counterpart, HadRM3P. Using a distributed computing infrastructure (Massey et al., 2006), very large ensemble of climate model simulations can be generated, allowing to examine rare extreme events with high statistical confidence.

Improvements in w@h2 include the use of a more recent land surface scheme, MOSES 2, which uses tiles to represent land surface types heterogeneity within each grid cell, as well as a two-fold increase in horizontal resolution in HadRM3P with the use of the $0.22°$ european CORDEX region. A large ensemble with about 100 members per year for years 1901–2006 has been generated, and is compared to a w@h1 ensemble over 1961–1990 (Massey et al., 2015).

Overall, w@h2 shows reduced biases compared to w@h1, although the general bias patterns persist. Biases in HadAM3P are reduced in the southern Hemisphere while mixed results are found in the northern hemisphere. The model is found to be reliable in most regions and in terms of year-to-year variability in global temperature over land. In HadRM3P, the most striking bias reduction is found over eastern Europe, where a warm summer bias is reduced (but remains significant). Precipitation biases in HadRM3P, on the other hand, do not exhibit substantial improvements overall. Hot extremes are overestimated for all European regions, but cold extremes are well represented. The model is shown to perform particularly well for extreme daily precipitation.

A limitation of w@h2 as presented in this study is the relatively short spin-up (1 year). We find that a longer spin-up may further improve w@h2, in particular with respect to the representation of summer temperatures over Southeastern Europe. Future w@h2 experiments will therefore include a longer spin-up of 5–10 years, in order to allow for a full stabilization of soil moisture and soil temperature and to thereby take full advantage of the capability of the model.

One of the main use of weather@home relates to the attribution of extreme weather events to anthropogenic climate change. The ability of the model to respond to forcing agents such as greenhouse gases and sea surface temperature was therefore examined over Europe. The model is reliable for seasonal averages of temperature, although slightly under-confident, i.e., it might underestimate the impact of the forcing. The model's reliability is less satisfactory for seasonally averaged precipitation, although in most regions and seasons comparison with observations lies within uncertainties.

Another common use of weather@home output is for the generation of data sets of synthetic extreme events, to be used by the impact modelling community. For example, the ongoing MaRIUS project (Managing the Risks, Impacts and Uncertainties of droughts and water Scarcity) uses drought events in the UK for present and future conditions generated by weather@home to assess the risks associated with droughts. Using the weather@home modelling system allows for thousands of drought events

to be generated and fed into various hydrological and impact models, thereby enabling a risk assessment framework to be applied to types of event with rather few observed occurrences.

For some applications, bias correction might be necessary. The availability of a large number of simulations allows for new methodologies to be applied, for example by re-sampling from the ensemble (Sippel et al., 2016b) or by giving weights to ensemble members in order to obtain distribution close to observations.

In this paper, we focused on the european region, but w@h2 is being developed over a range of regions. Collaborators around the world have already used weather@home, where HadRM3P is run over their region of interest, and the project is expected to continue establishing new regions with w@h2 in the future.

In conclusion, the improved physical representation of the land surface in w@h2 increases our confidence in the model's ability to simulate weather extremes, in particular hot extremes which can be highly related to land surface-atmosphere interactions (e.g., Miralles et al., 2014), although some biases persist. Overall, weather@home may be a useful tool for the investigation of extreme weather events if proper bias corrections and other caveats are taken into account.

## 6 Code and availability

HadRM3P is available from the UK Met Office as part of the Providing REgional Climates for Impacts Studies (PRECIS) program. Access to standard versions of the software is dependent on attendance at a PRECIS training workshop after which all source code, including that relevant to configuring HadAM3P, and other materials is made available (http://www.metoffice.gov.uk/research/applied/international-development/precis/obtain). These workshops are either held at the Met Office, for which a small charge is made to cover costs of the workshop delivery, or as part of a project, often in a region where PRECIS is to be applied. The code to manage and embed these models within the weather@home project is specific to their utilisation within the BOINC environment and we consider not within the scope of this publication.

The full set of model output data for the experiment used in this study will be freely available on the Centre for Environmental Data Analysis (http://www.ceda.ac.uk) in the next few months. Until point of publication within the CEDA archive please email cpdn@oerc.ox.ac.uk who will work with you to access the relevant data.

*Author contributions.* The model simulations were designed by B.P. Guillod with input from R.G. Jones, M.R. Allen and F.E.L. Otto. All results were analysed and plotted by B.P. Guillod. The paper was written by B.P. Guillod, with edits from all co-authors. The weather@home2 model code was configured for and ported to the BOINC infrastructure by S. Wilson, S.N. Sparrow, A. Bowery and D. Wallom. Testing was done by B.P. Guillod, N.R. Massey and D.M. Mitchell.

*Acknowledgements.* This work was undertaken within the MaRIUS project: Managing the Risks, Impacts and Uncertainties of droughts and water Scarcity, funded by the Natural Environment Research Council (NERC), and undertaken by a project team spanning the University of Oxford [NE/L010364/1], University of Bristol [NE/L010399/1], Cranfield University [NE/L010186/1], the Met Office, and the Centre for Ecology and Hydrology [NE/L010208/1]. The integration of TRIFFID into weather@home was supported by a grant from the USDA

National Institute of Food and Agriculture (2013-67003-20652). We would also like to thank the Met Office Hadley Centre PRECIS team for their technical and scientific support for the development and application of weather@home. Finally, we would like to thank all of the volunteers who have donated their computing time to climateprediction.net and weather@home. We acknowledge the E-OBS dataset from the EU-FP6 project ENSEMBLES (http://ensembles-eu.metoffice.com) and the data providers in the ECA&D project (http://www.ecad.eu).

5   CRU-TS data was downloaded from https://crudata.uea.ac.uk/cru/data/hrg and JRA-55 from the ECMWF website. The SRB dataset was obtained from the NASA Langley Research Center Atmospheric Science Data Center and the FLUXNET-MTE dataset was downloaded from http://www.bgc-jena.mpg.de/geodb/BGI/Home. We are grateful to CEDA (Centre for Environmental Data Analysis, NERC) and their Jasmin analysis platform (Lawrence et al., 2013) on which data analysis has been done. Finally, we thank the two anonymous reviewers, whose comments have helped to improve the manuscript.

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

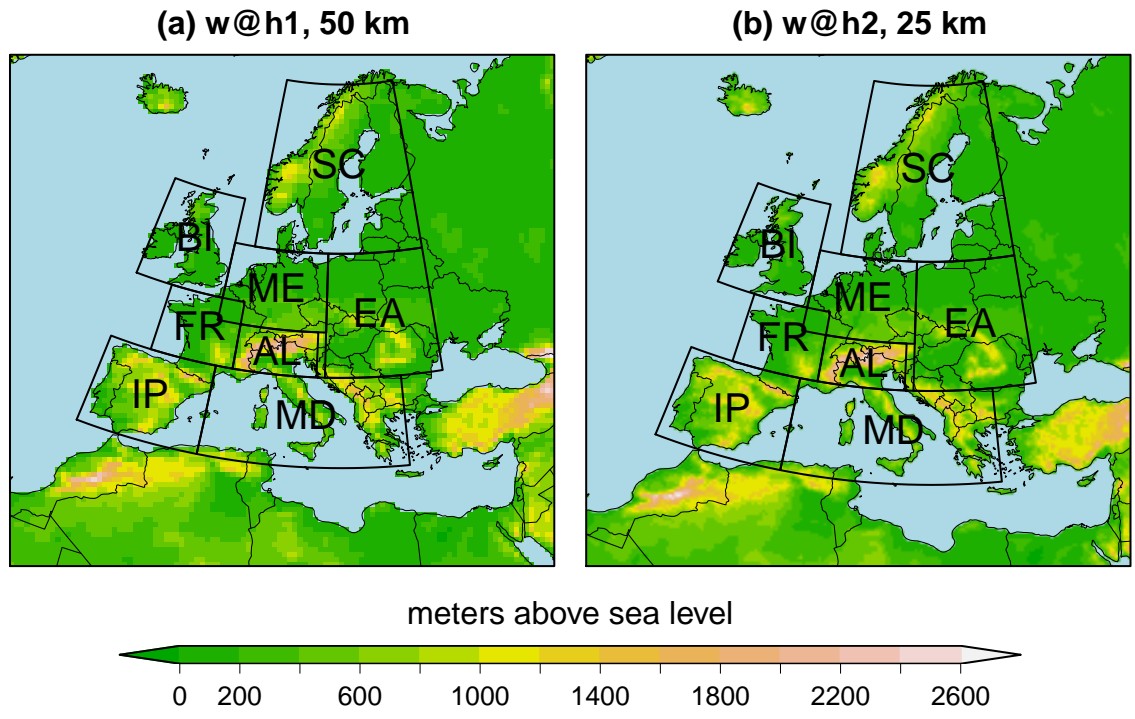

**Figure 1.** HadRM3P domain (excluding sponge layer) and topography with the subdomains used in the analysis. (a) The w@h1, 0.44° domain, (b) the w@h2, 0.22° domain. The subdomains are those defined in the PRUDENCE project (Christensen and Christensen, 2007): the Alps (AL), the British Isles (BI), Eastern Europe (EA), France (FR), the Iberian Peninsula (IP), the Mediterranean (MD), Mid-Europe (ME), and Scandinavia (SC).

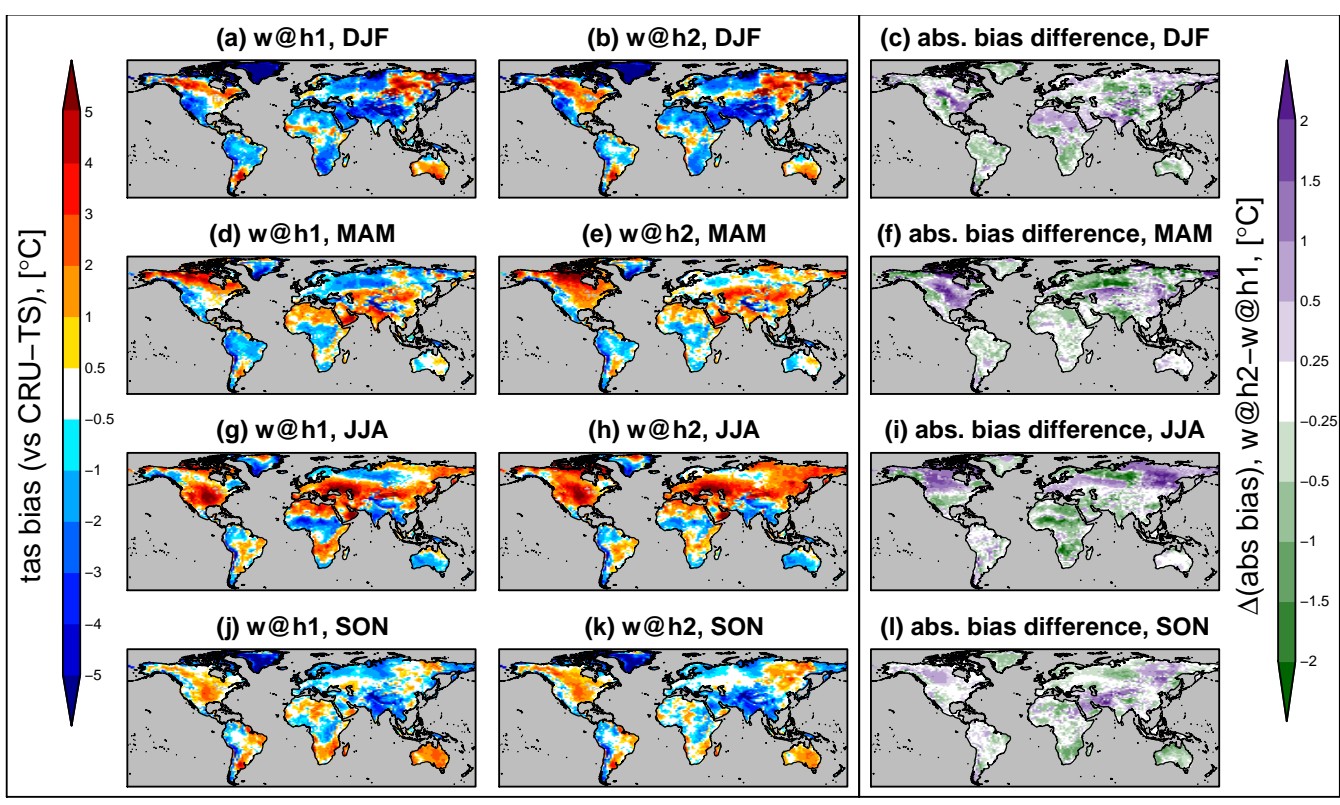

**Figure 2.** Biases in surface air temperature for the GCM HadAM3P in w@h1 (left; a,d,g,j) and w@h2 (middle; b,e,h,k), and the difference in absolute biases (right; c,f,i,l, expressed as w@h2 minus w@h1, i.e., negative values indicate an improvement). Each row corresponds to a season (from top to bottom: DJF, MAM, JJA, SON). Biases are computed with respect to the CRU-TS dataset and are expressed in °C, and grey indicates regions without data (ocean grid cells).

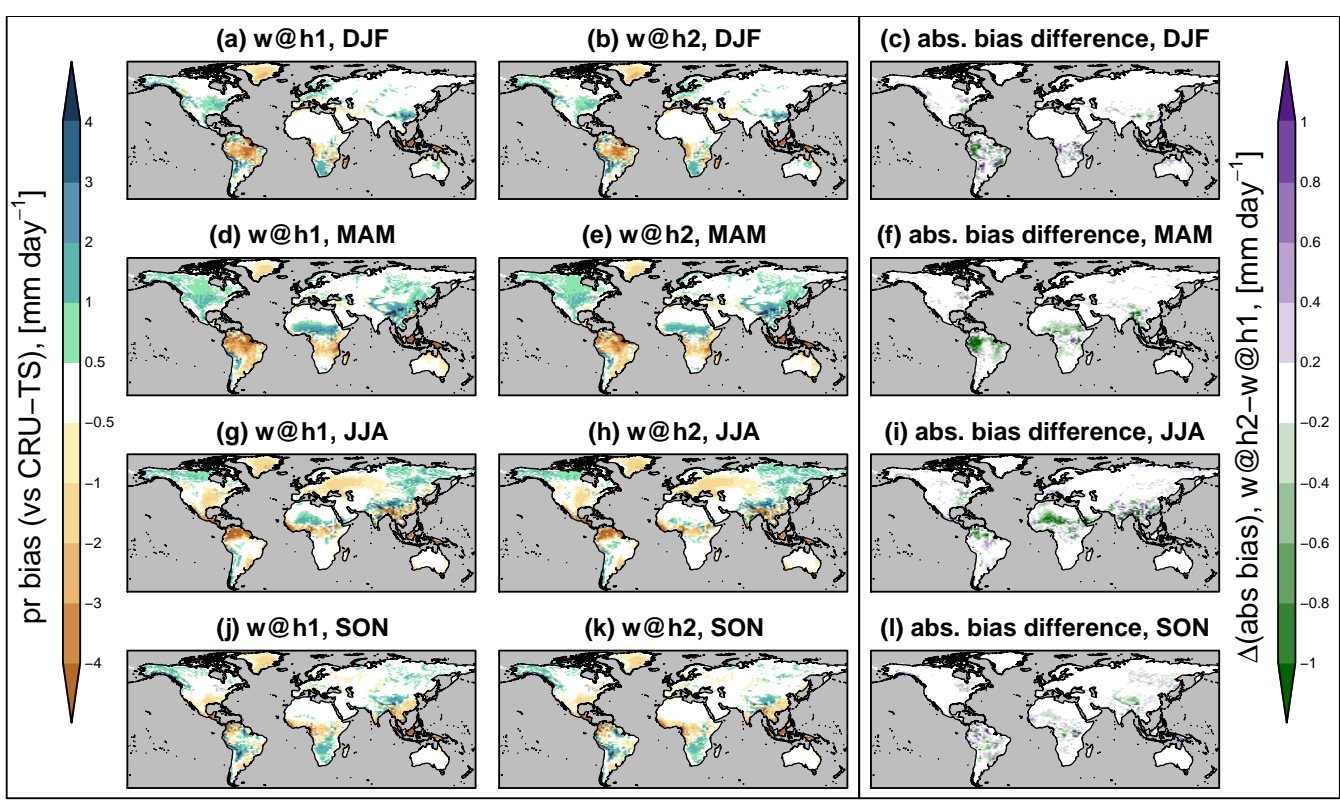

**Figure 3.** Same as Fig. 2 but for precipitation, in $\mathrm{mm\,day^{-1}}$.

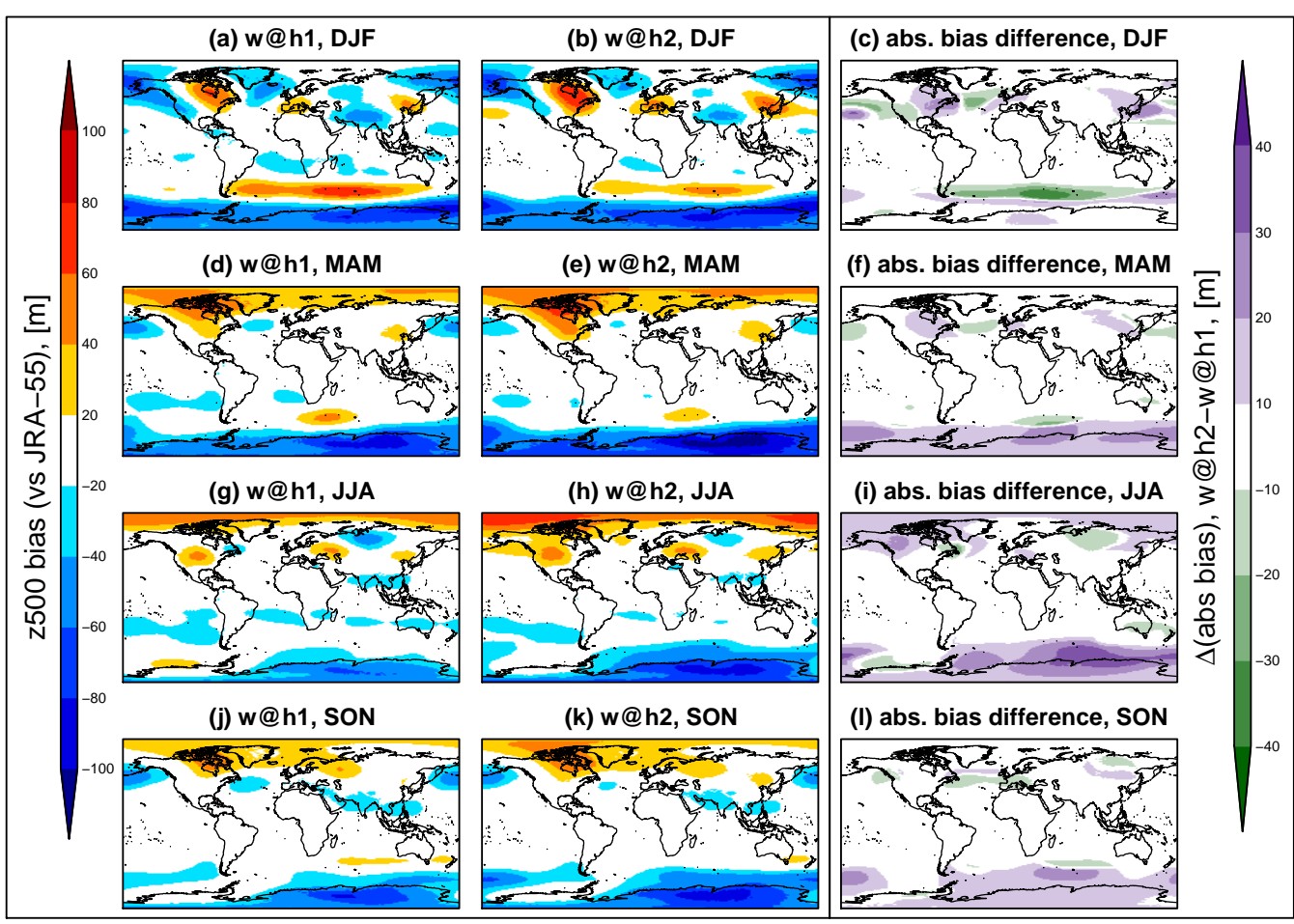

**Figure 4.** Same as Fig. 2 but for geopotential height at 500 hPa with respect to the JRA-55 reanalysis.

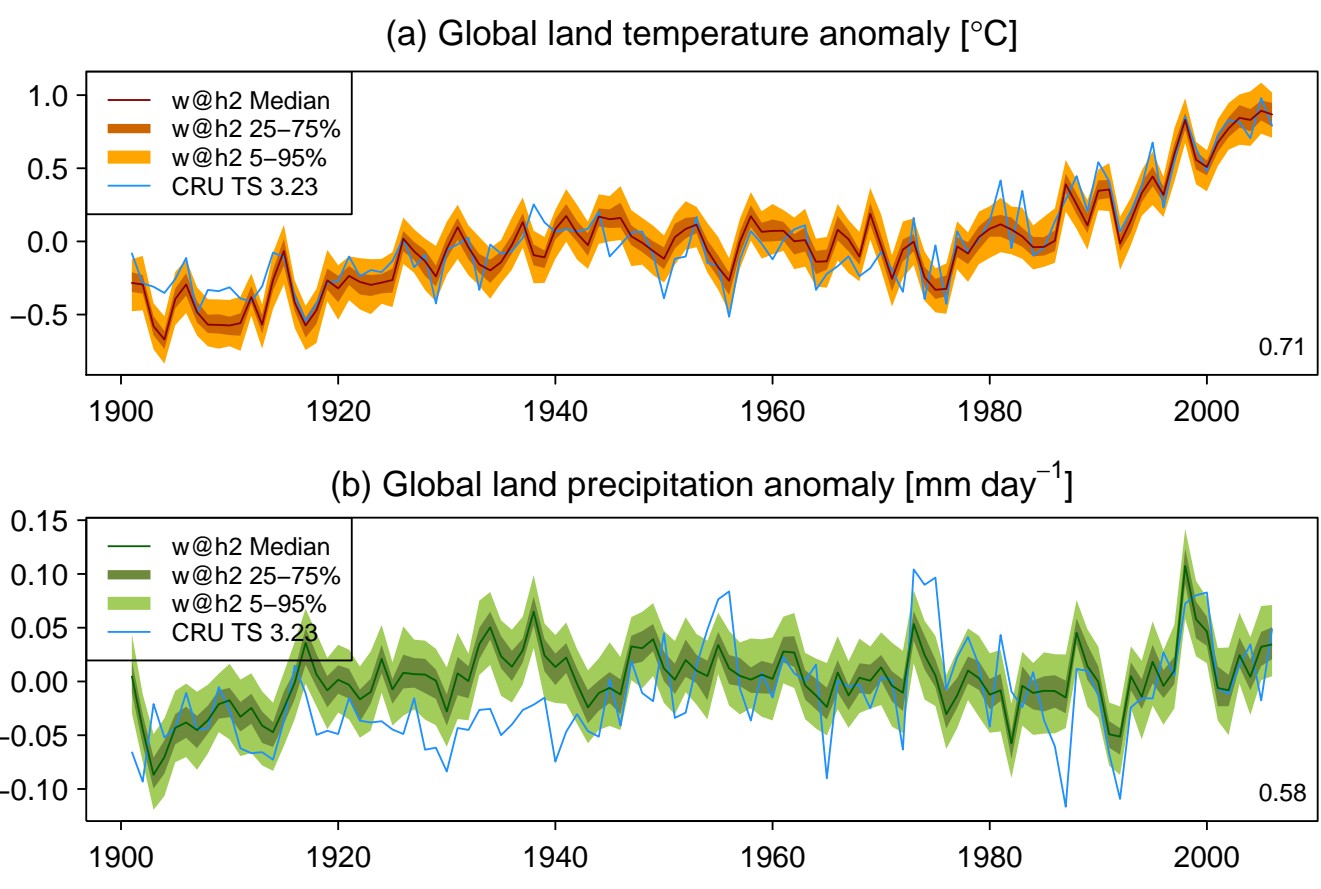

**Figure 5.** Global land annual mean time series of (a) temperature and (b) precipitation in HadAM3P in w@h2 and CRU-TS, expressed as anomalies relative to 1961–1990. The median, inter-quartile range, and 5–95% range of the w@h2 ensemble members are shown for each year. Antarctica is not included, as in CRU-TS. The fraction of years with observed value lying outside of the 5–95% range of the w@h2 ensemble members is shown in the lower right of each plot. Time series with actual values (i.e., not anomalies) are shown in Supplementary Fig. S8.

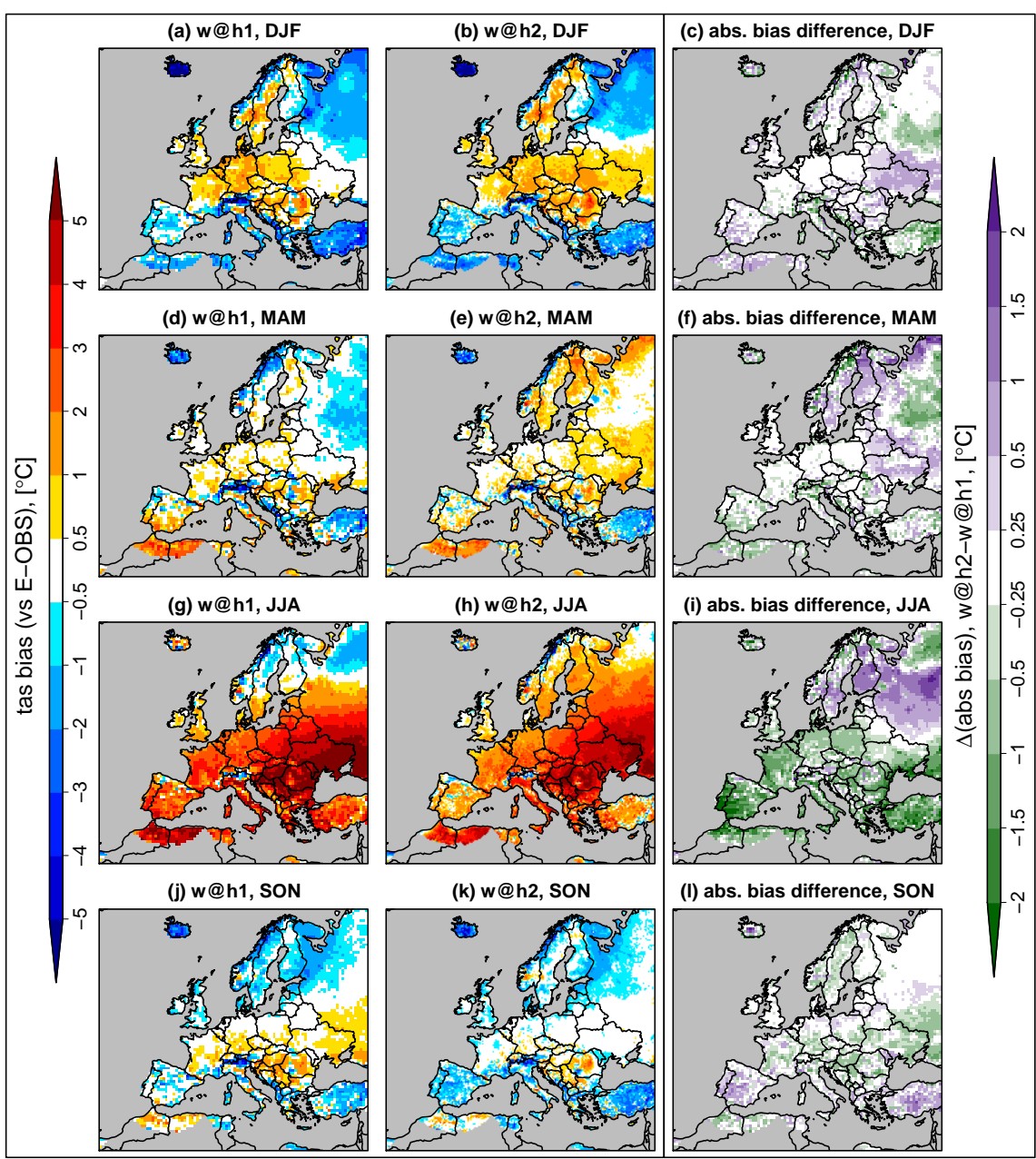

**Figure 6.** Biases in surface air temperature for the RCM HadRM3P in w@h1 (left; a,d,g,j) and w@h2 (middle; b,e,h,k), and the difference in absolute biases (right; c,f,i,l, expressed as w@h2 minus w@h1, i.e., negative values indicate an improvement). Each row corresponds to a season (from top to bottom: DJF, MAM, JJA, SON). The biases are computed on the respective model resolution, while the absolute bias difference is computed on the 0.44° grid. Biases are computed with respect to the E-OBS dataset and are expressed in °C.

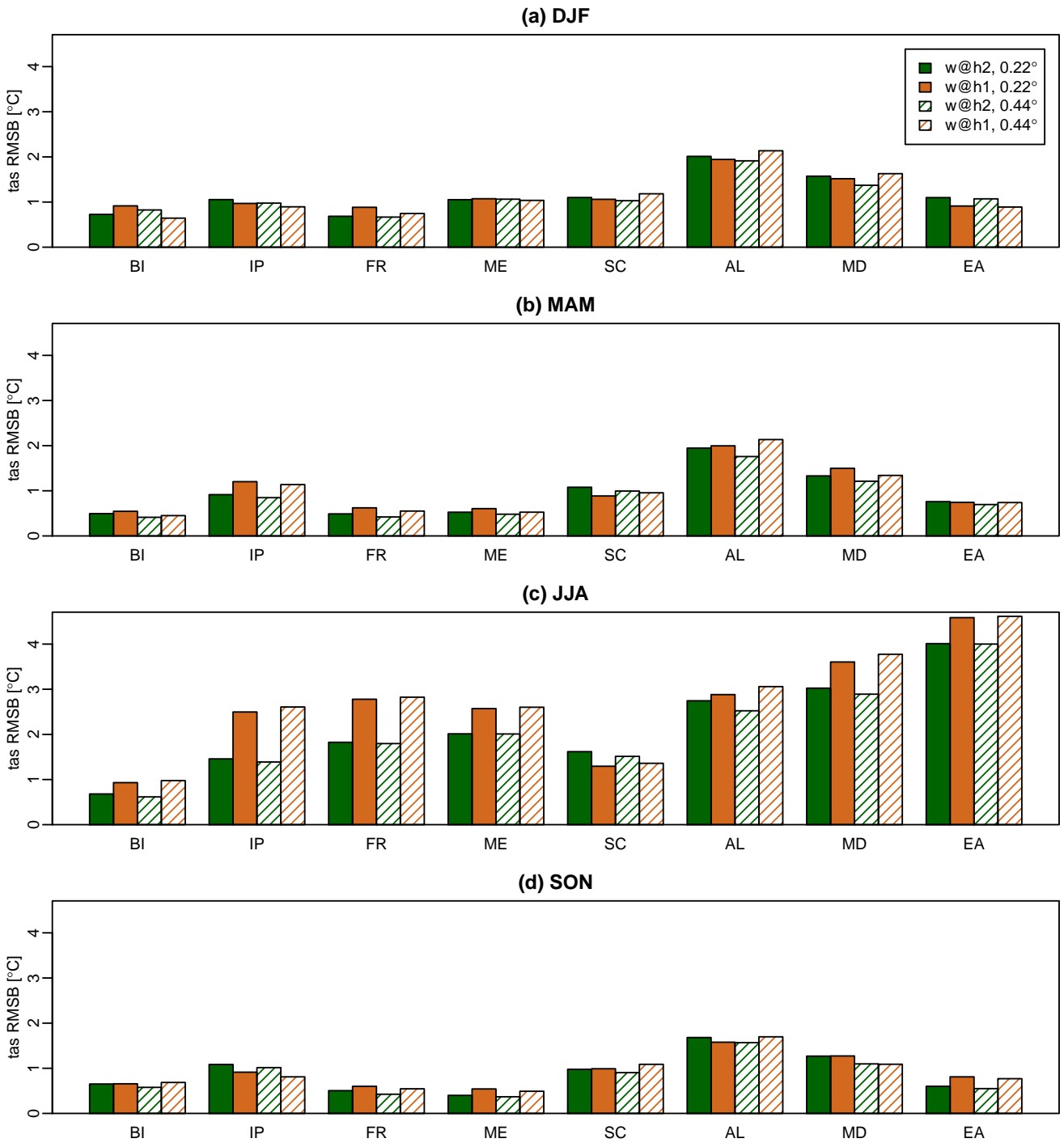

**Figure 7.** Spatial root-mean squared biases in surface air temperature by region and season with respect to the E-OBS dataset. The regions are shown in Fig. 1. Green (orange) bars are for w@h2 (w@h1), filled (hatched) bars are in comparison with E-OBS at 0.22° (0.44°) resolution (for w@h1, 0.22 is done by bilinear interpolation; for w@h2, 0.44 is done by aggregation).

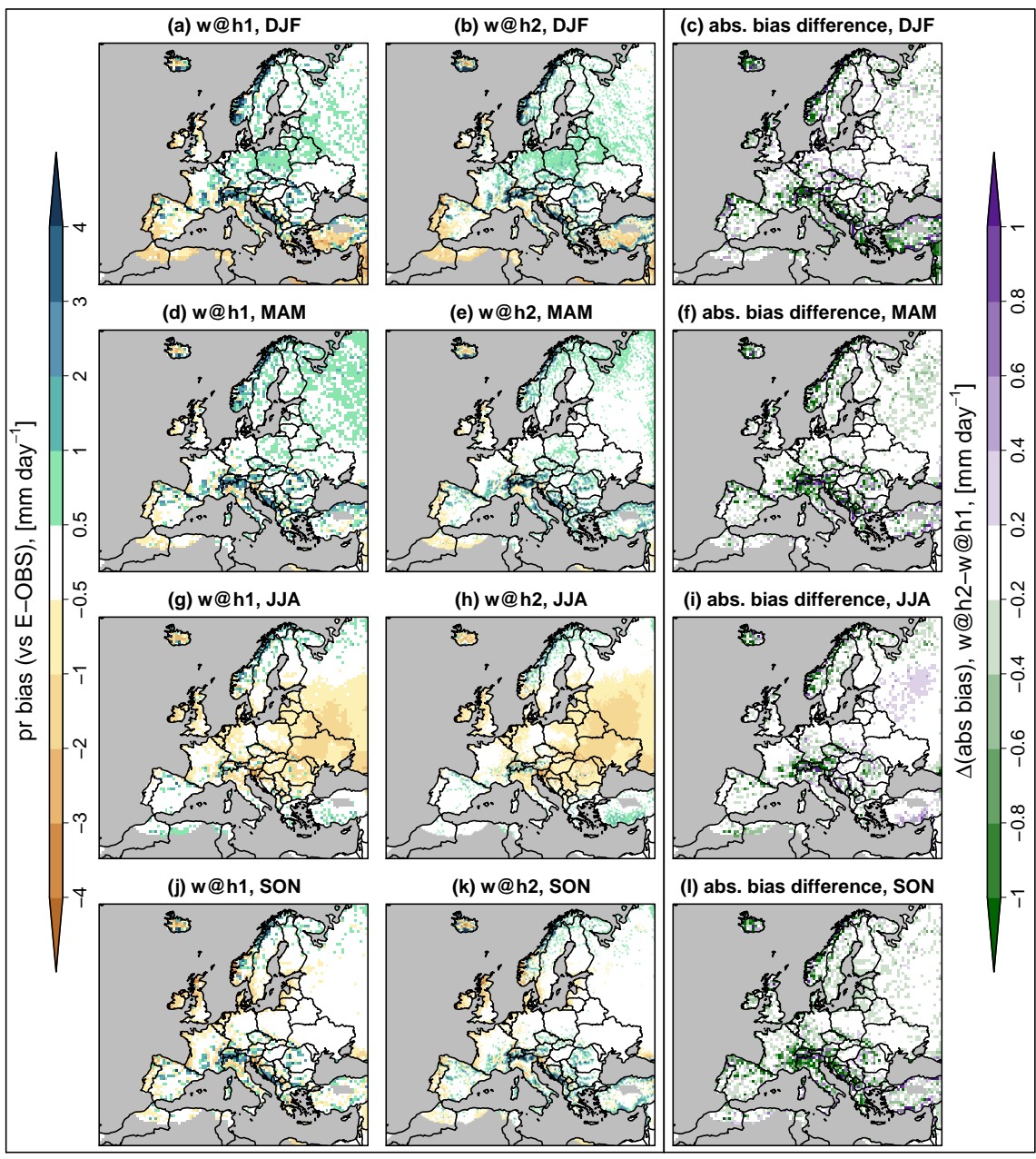

**Figure 8.** Same as Fig. 6 but for precipitation, in $\mathrm{mm\,day^{-1}}$. See Supplementary Fig. S13 for these biases in relative terms.

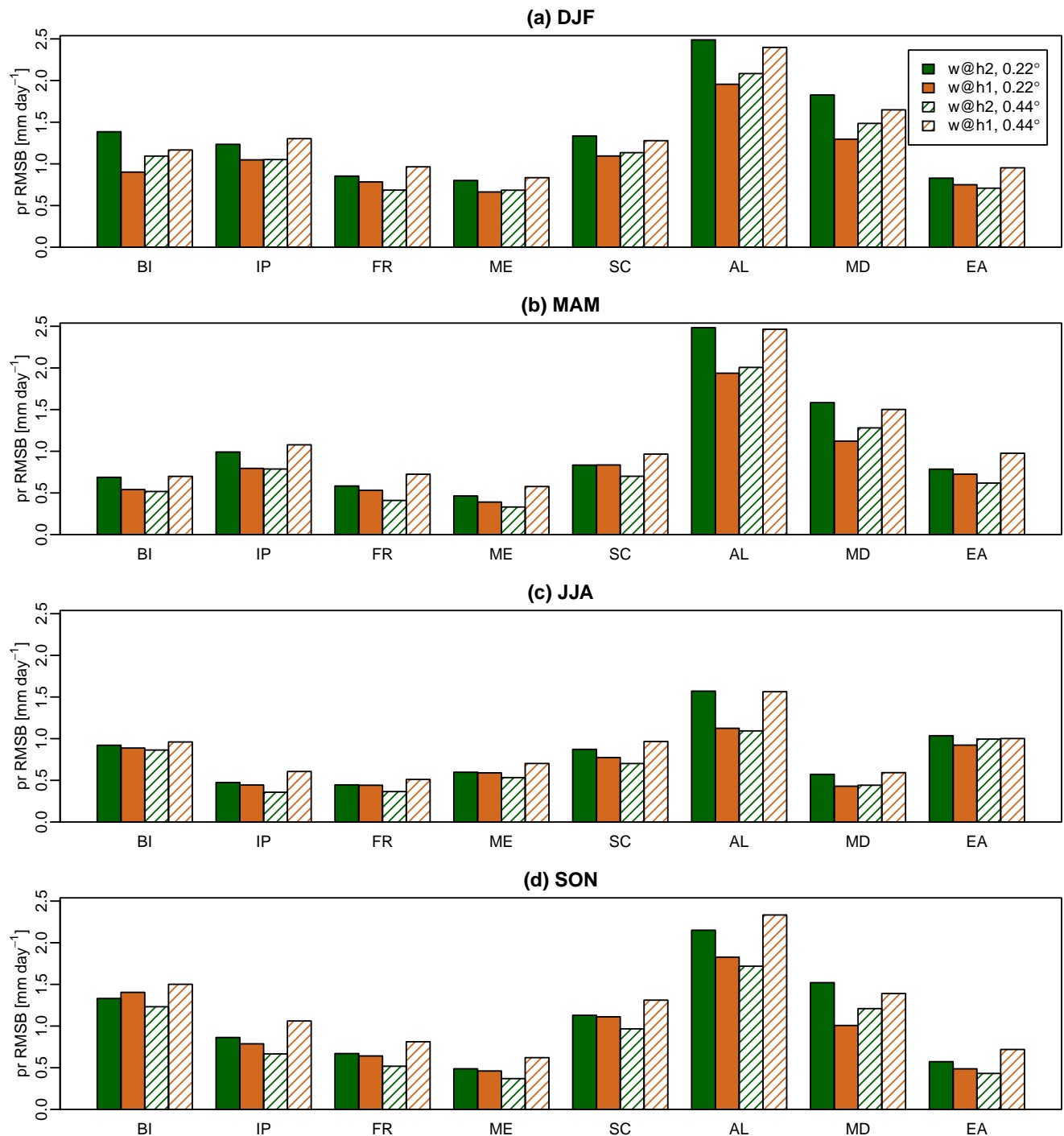

**Figure 9.** Same as Fig. 7 but for precipitation.

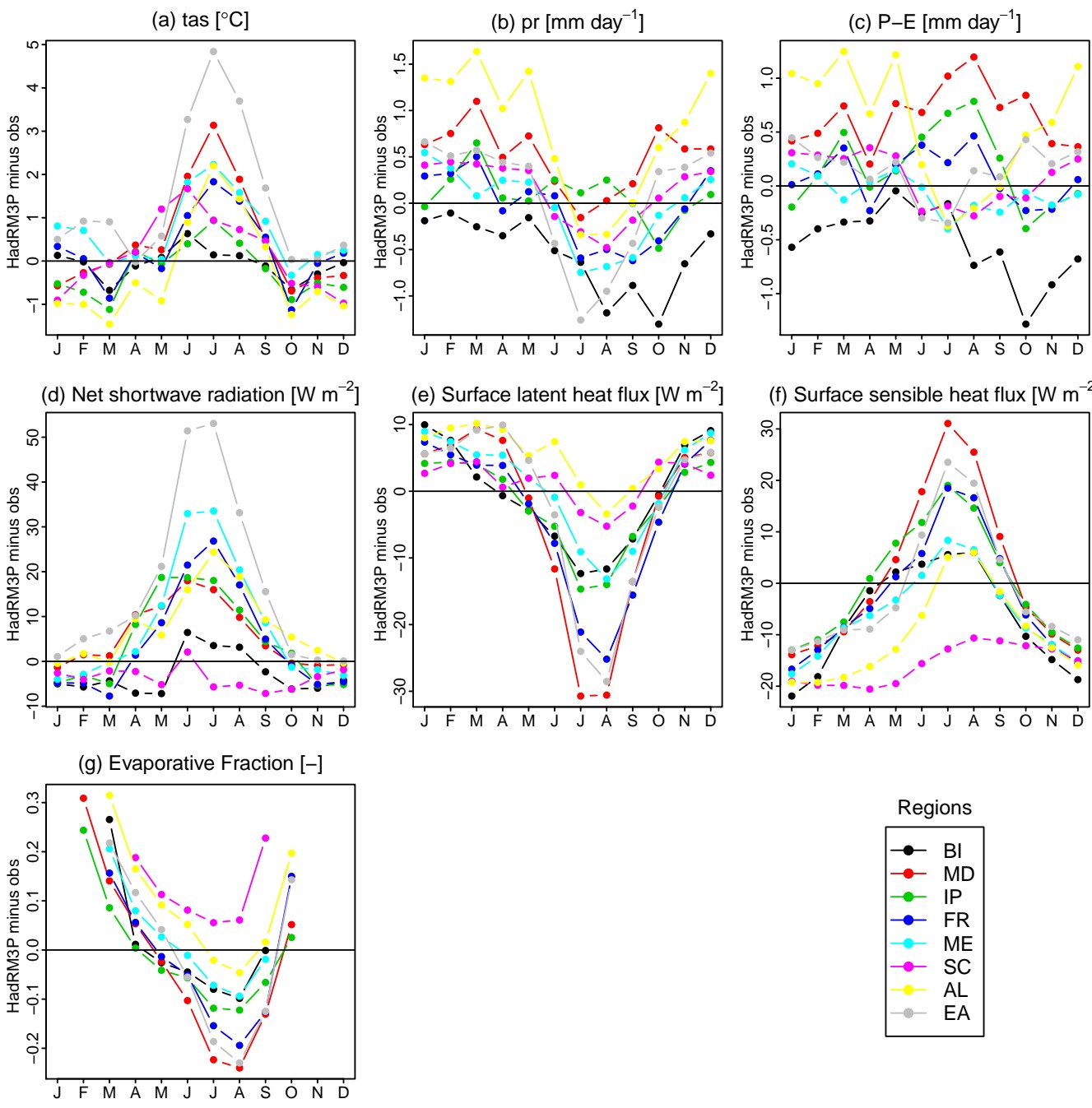

**Figure 10.** Seasonal cycle of HadRM3P biases from 1984–2006 in each region for (a) temperature, (b) precipitation, (c) precipitation minus evaporation, (d) net shortwave radiation at the surface, (e) surface latent heat flux, (f) surface sensible heat flux and (g) evaporative fraction (i.e., the ratio of latent heat flux to the sum of sensible and latent heat fluxes). The observational dataset used are E-OBS for temperature and precipitation, SRB for radiation, FLUXNET-MTE for surface fluxes and the evaporative fraction. P-E is computed using precipitation from E-OBS and evapotranspiration from FLUXNET-MTE.

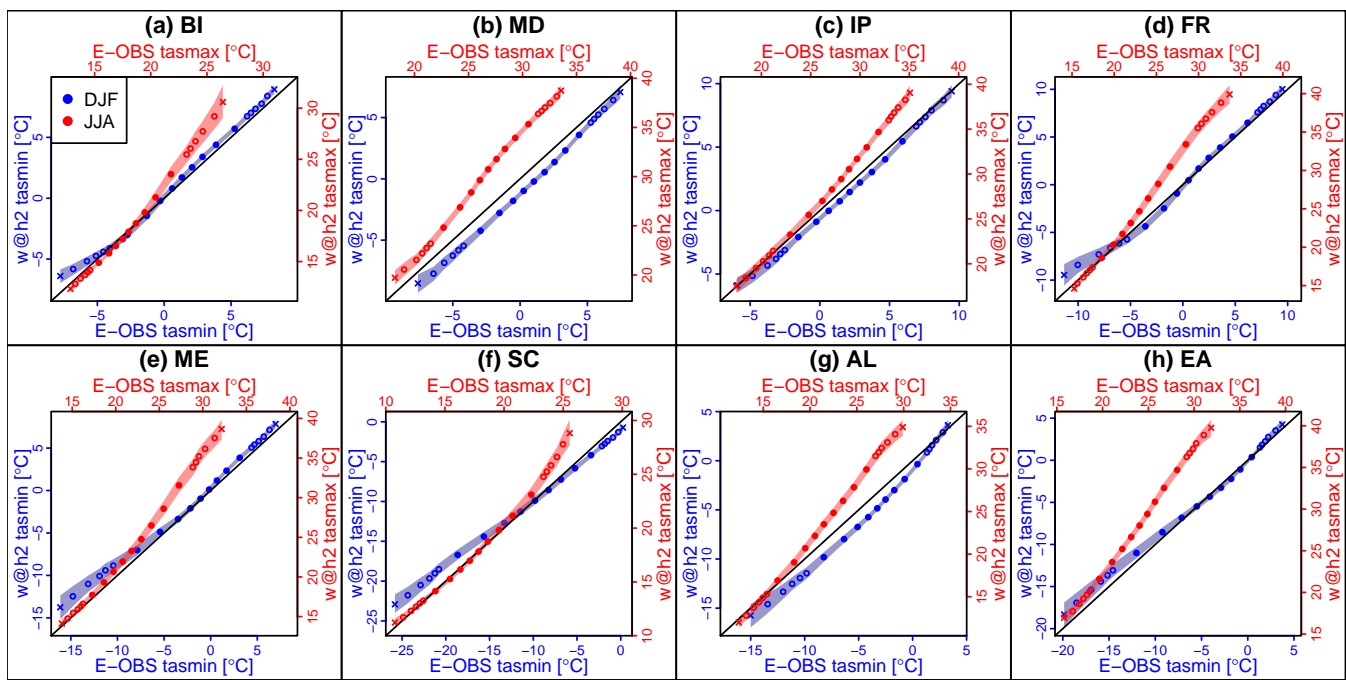

**Figure 11.** Quantile-quantile plots of the distribution of (red) JJA daily maximum temperature (tasmax) and (blue) DJF daily minimum temperature (tasmin) for the 8 regions, comparing w@h2 to E-OBS over years 1950-2006. Blue axes (bottom, left) are for DJF tasmin, red axes (top, right) are for JJA tasmax. Dots show the quantile values for the entire ensemble (filled dots: deciles; empty dots: 1st to 5th and 95th to 99th percentiles; cross symbols: 0.5 and 99.5 percentile). The coloured envelopes show the 95% confidence interval of w@h2 quantile values from 1000 bootstrap samples with one ensemble member per year (see text).

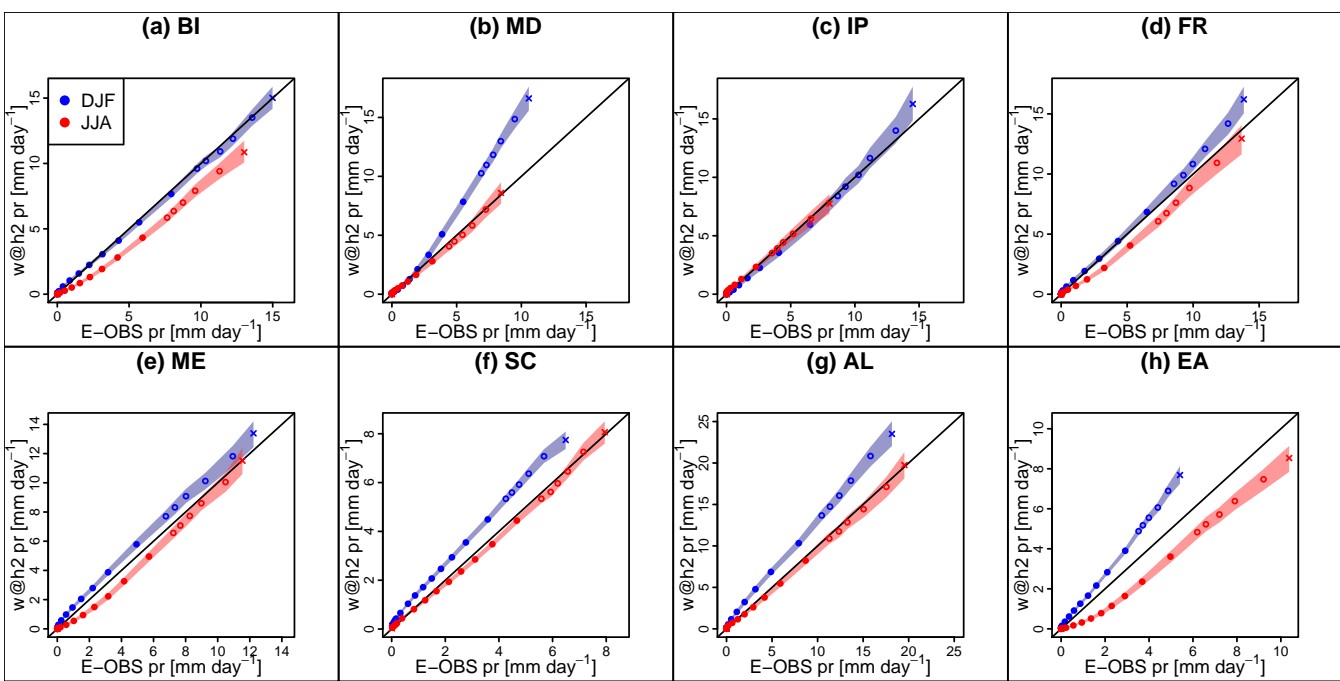

**Figure 12.** Same as Fig. 11 but for daily precipitation: Quantile-quantile plots for JJA (red) and DJF (blue) comparing w@h2 to E-OBS over years 1950-2006. Here, the same axes are used for both seasons.

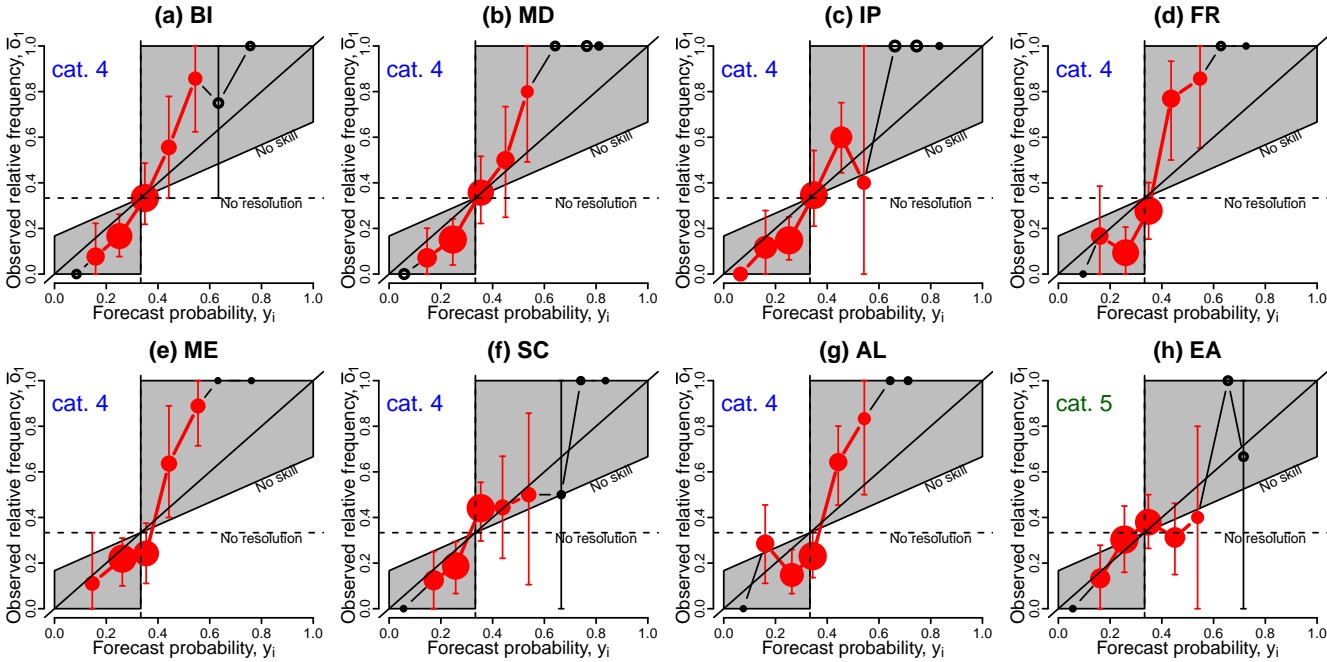

**Figure 13.** Reliability diagrams for high summer temperature for each sub-region, defined as seasonally averaged JJA temperature in the upper tercile over 1902–2006. Forecasts are grouped into bins of size 0.1 and their average value within each bin is plotted (x-axis) versus the relative frequency observed on the corresponding years. The area of the dots is proportional to the number of forecasts within each bin, with bins containing less than five years shown in black (red dots indicate bins containing at least 5 years). Error bars are computed from 100 bootstrap samples using the R package "verification" (NCAR – Research Applications Laboratory, 2015). Grey shading indicates where data points contribute positively to skill (Wilks, 2011). Performance category is indicated in the upper left of each plot, on a scale from 1 (dangerously useless) to 5 (perfect) (see Table 3) as in Weisheimer and Palmer (2014).

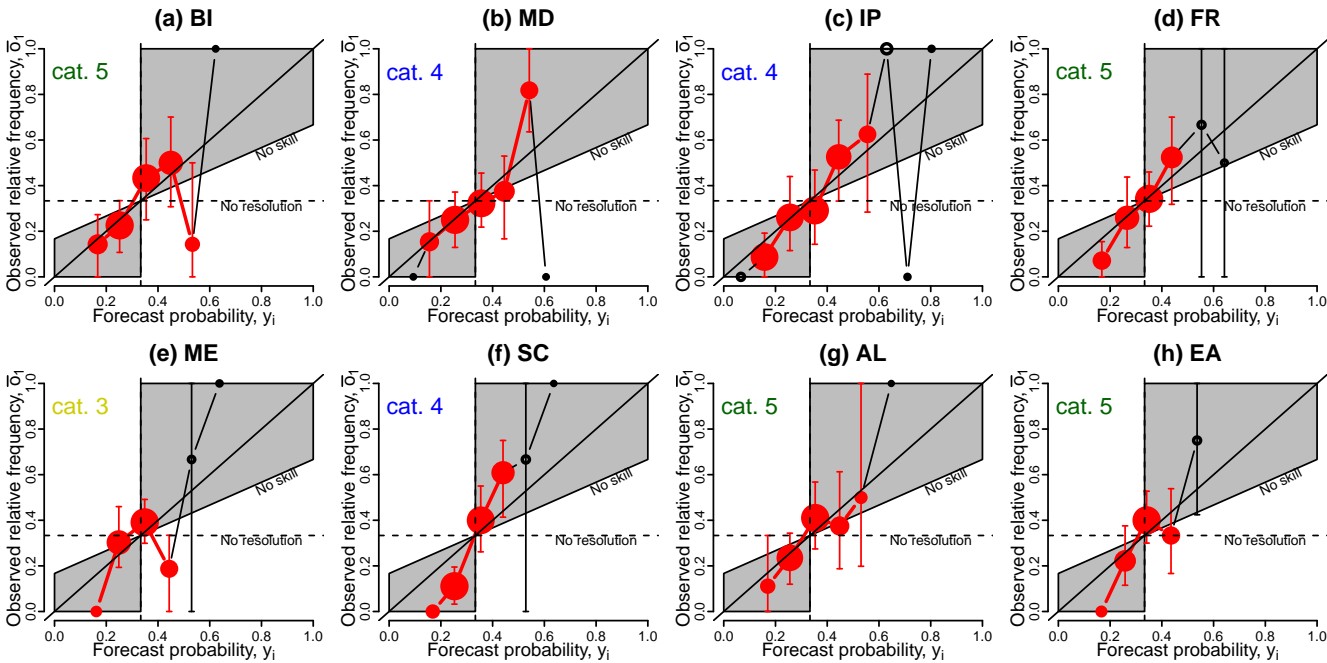

**Figure 14.** Reliability diagrams for low winter temperature, defined as seasonal DJF averages in the lower tercile. See the caption of Fig. 13 for technical details.

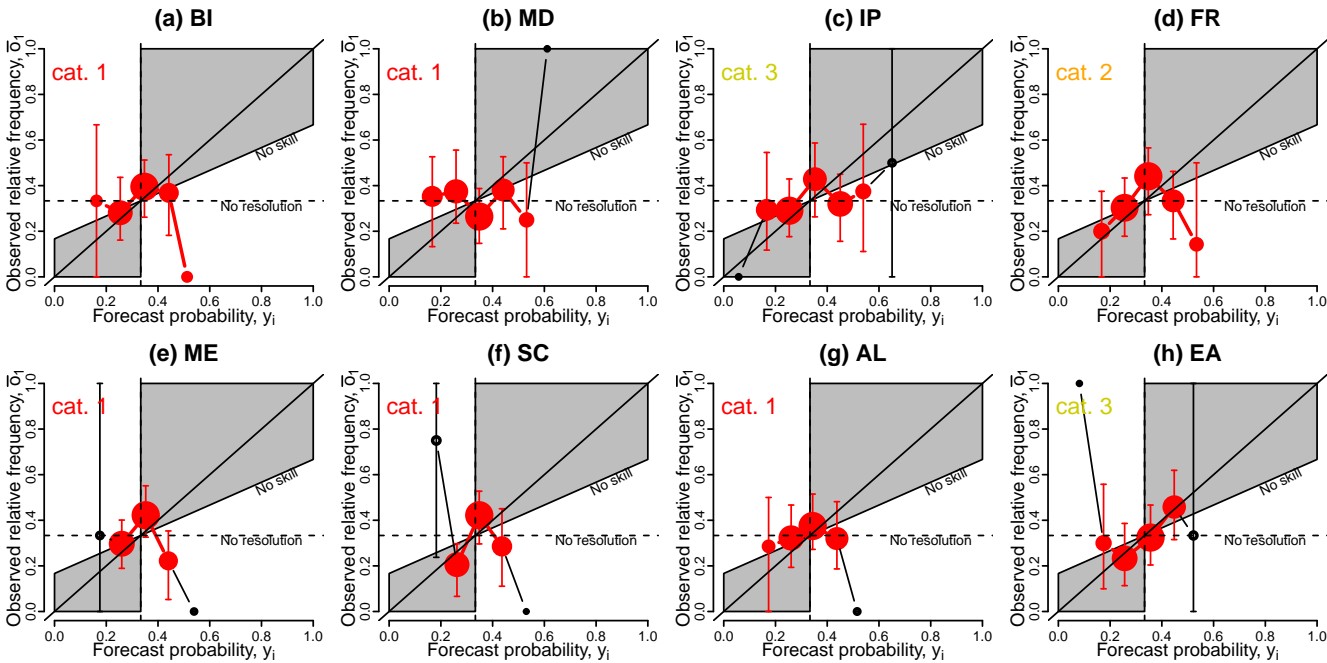

**Figure 15.** Reliability diagrams for low summer precipitation, defined as seasonal JJA averages in the lower tercile. See the caption of Fig. 13 for technical details.

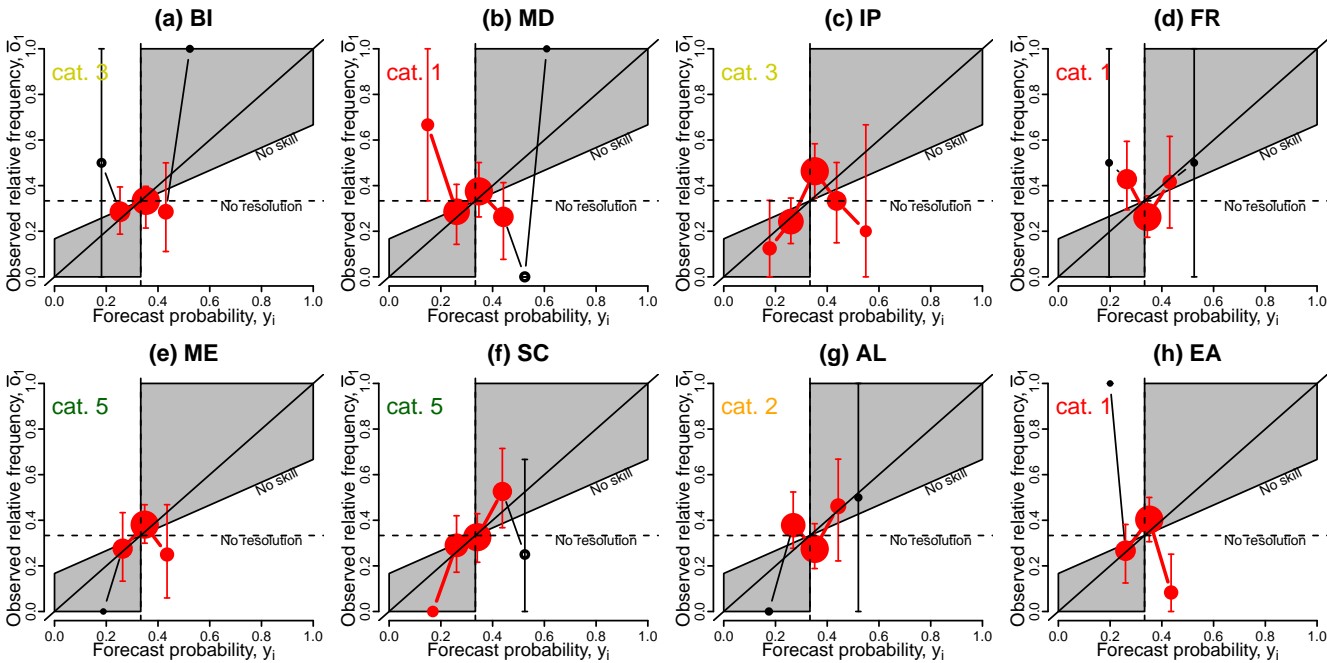

**Figure 16.** Reliability diagrams for high winter precipitation, defined as seasonal DJF averages in the upper tercile. See the caption of Fig. 13 for technical details.

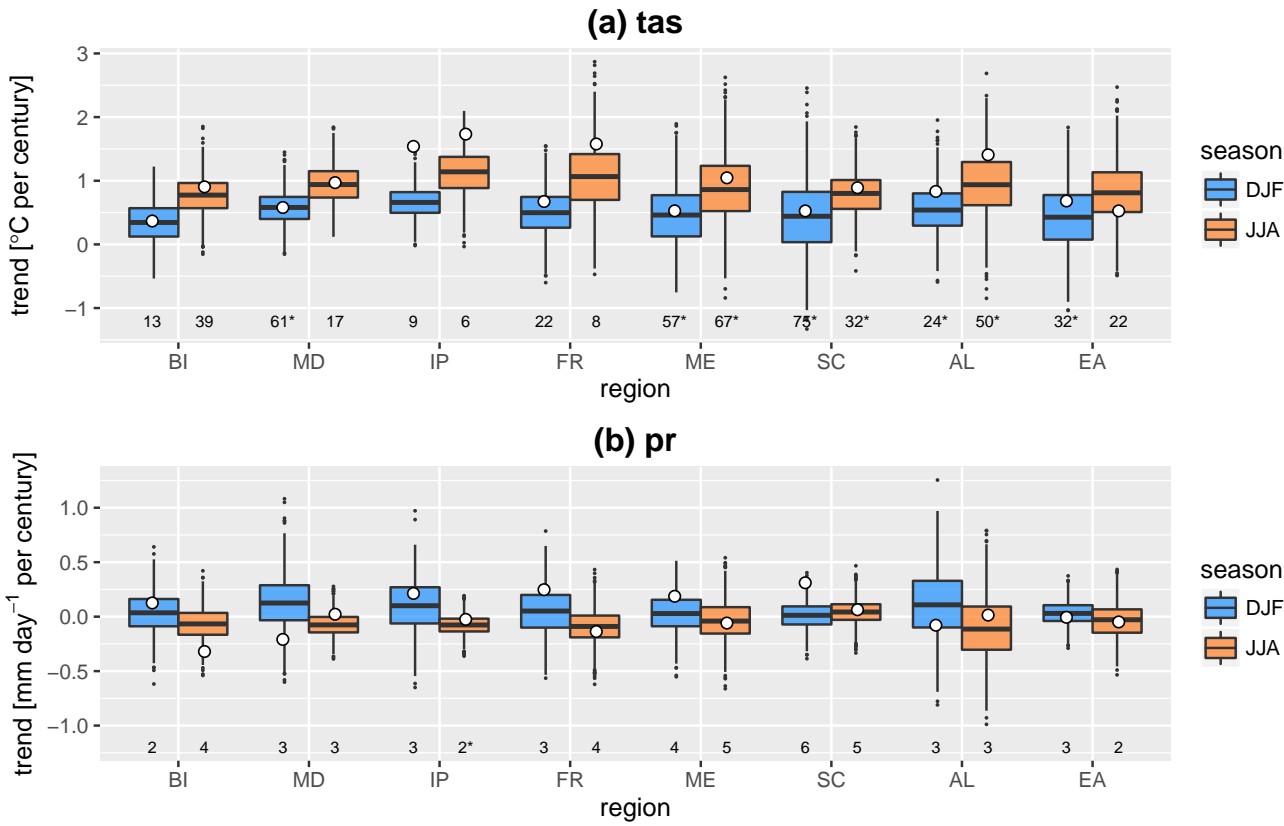

**Figure 17.** Regional summer and winter trends in (a) temperature and (b) precipitation. Boxes show the distribution of trend values using 1000 time series constructed by randomly sampling one w@h2 ensemble member per year, with outliers shown as black dots. White dots show the observed regional trend estimated from CRU-TS. Theil-Sen linear trend slope are computed using regional averages and significance is tested using a Mann-Kendall test. The numbers below the boxes indicate the percentage of w@h2 time series with a statistically significant trend (at the 5% level), with an asterisk if the observed trend is significant.

**Table 1.** Global root mean squared biases by season for HadAM3P in w@h1 and w@h2. Individual grid cells are weighted by their area.

|  |  | DJF | MAM | JJA | SON |
|---|---|---|---|---|---|
| Air temperature (°C) | w@h1 | 3.01 | 1.78 | 2.25 | 1.84 |
|  | w@h2 | 2.99 | 1.76 | 2.22 | 1.77 |
| Precipitation (mm day$^{-1}$) | w@h1 | 1.47 | 1.72 | 1.81 | 1.45 |
|  | w@h2 | 1.47 | 1.61 | 1.75 | 1.43 |

**Table 2.** HadRM3P biases in regionally averaged temperature and precipitation for the 8 regions shown in Fig. 1. The numbers in bold font indicate better performance in the corresponding model version for each region, season and variable.

|  |  |  | BI | IP | FR | ME | SC | AL | MD | EA |
|---|---|---|---|---|---|---|---|---|---|---|
| Air temperature (°C) | DJF | w@h1 | **0.20** | **-0.51** | 0.56 | 0.99 | -0.19 | -0.92 | -1.04 | **0.65** |
|  |  | w@h2 | 0.37 | -0.67 | **0.45** | **0.98** | **0.06** | **-0.75** | **-0.67** | 0.93 |
|  | MAM | w@h1 | 0.10 | 0.41 | 0.35 | 0.41 | **-0.23** | -0.87 | 0.12 | **0.26** |
|  |  | w@h2 | **0.01** | **0.22** | **0.20** | **0.33** | 0.47 | **-0.62** | 0.12 | 0.47 |
|  | JJA | w@h1 | 0.75 | 2.26 | 2.74 | 2.55 | **0.30** | 2.07 | 3.50 | 4.48 |
|  |  | w@h2 | **0.35** | **0.98** | **1.72** | **1.93** | 0.98 | **1.76** | **2.66** | **3.89** |
|  | SON | w@h1 | -0.52 | **-0.25** | 0.25 | 0.34 | -0.84 | **-0.62** | **-0.31** | 0.50 |
|  |  | w@h2 | **-0.40** | -0.71 | **-0.2** | **-0.02** | **-0.61** | -0.76 | -0.41 | **0.05** |
| Precipitation (mm day$^{-1}$) | DJF | w@h1 | -0.31 | -0.43 | **0.07** | **0.47** | **0.58** | **0.99** | **0.03** | **0.43** |
|  |  | w@h2 | **0.04** | **-0.42** | 0.17 | 0.51 | 0.64 | 1.01 | 0.34 | 0.47 |
|  | MAM | w@h1 | -0.26 | **0.04** | 0.07 | 0.14 | 0.52 | **1.08** | **0.60** | 0.46 |
|  |  | w@h2 | **-0.19** | 0.07 | 0.07 | **0.11** | 0.44 | 1.19 | 0.74 | **0.44** |
|  | JJA | w@h1 | -0.78 | 0.14 | -0.37 | -0.48 | **-0.10** | -0.40 | **-0.07** | **-0.84** |
|  |  | w@h2 | **-0.73** | **0.09** | **-0.34** | **-0.46** | -0.13 | **-0.20** | -0.09 | -0.93 |
|  | SON | w@h1 | -1.19 | -0.09 | -0.28 | -0.23 | -0.24 | **0.71** | **0.19** | **0.02** |
|  |  | w@h2 | **-0.90** | **-0.06** | **-0.20** | **-0.11** | **-0.05** | 0.77 | 0.50 | 0.18 |

**Table 3.** Characterization of forecasts reliability following Weisheimer and Palmer (2014). The uncertainty range of the slope is characterised by the 75% confidence interval derived from 1000 bootstrap samples with replacement.

| Category | Meaning | Slope of the reliability diagram |
|---|---|---|
| 5 | perfect forecast | Uncertainty range includes perfect reliability (i.e., 1) |
| 4 | still very useful for decision-making | Lower uncertainty bound is at minimum of 0.5 and uncertainty range does not include the perfect reliability line. |
| 3 | marginally useful | Lower uncertainty bound is positive but does not belong to category 4 or 5. |
| 2 | not useful | Positive slope and uncertainty range includes 0. |
| 1 | dangerously useless | Negative slope. |