# Peer review of "weather@home 2: validation of an improved global-regional climate modelling system"

_Geoscientific Model Development, 2016_

## Referee Comment (RC1) · Anonymous Referee #1 · 23 Nov 2016

The manuscript describes the validation of the improved global-regional climate modeling system weather@home2. Reading this manuscript has been a real pleasure! The manuscript describes in a concise and very well written way the changes compared to the previos version of the model system and their impact on the global and regional climate over Europe. Figures illustrate the important results. I can recommend this manuscript for publication after the authors have addressed a few questions and comments.

Abstract and Conclusions: I wouldn't fully agree to the statement that European biases are reduced. It is certainly true for the temperature, but precipitation? Look at Fig.9, the w@h1 0.22 deg results are often better than w@h2! I suggest you differentiate between temperature and precipitation biases.

[Figure]

Sec2.3: it is not clear to me how you create the initial conditions for each simulation. Are single-year spinup simulations part of the 13-mnth long experiment (i.e. making it 25 month long), or how exactly is it done? Please explain.

Sec2.3: How good is the initialization of soil and vegetation variables? Soil has a memory in excess of 1 year, so a 1-yr spin-up may not be sufficient for soil temperature and humidity. You have made a large effort to improve the land surface and vegetation components in your model, yet an inaccurate initialization could make these improvements worthless. Could you comment on that?

Sec4.4: To be honest, I was somewhat surprised to see a section about reliability in this manuscript. Reliability is a very specific term with a precise definition in the verification of probabilistic forecasts, but I have never encountered it in the context of climate simulations. On the other hand, the reliability of climate models is often discussed (e.g. in the IPCC AR) in the casual meaning of reliability as a synonym to trustworthiness. In this second definition of reliability, one often looks at how well the pdf of a quantity from a climate model matches the observed distribution. I wonder if this latter approach was what you had in mind when you started discussing reliability. Reliability and attribution diagrams as you present them now don't make much sense in the context of climate simulations, they should only be used for the verification of probabilistic forecasts. I therefore suggest you remove section 4.4 completely.

---

## Referee Comment (RC2) · Anonymous Referee #2 · 27 Jan 2017

Review of 'weather@home 2: validation of an improved global-regional climate modelling system' by Guillod et al.

The paper is a useful and fairly thorough documentation of the w@h system with a focus on Europe, although some gaps remain that can easily be filled. Also, there are some qualitative statements that can be converted to quantitative ones with, I think, relatively minor effort.

I am looking forward to a more complete version, which would be very informative.

Major comments

1. Can you comment how the biases in the global model compare to other, state-of-the-art, GCMs, eg in Chapter 9 of the IPCC WG1 AR5?

2. For attribution, a correct representation of variance is as important as trends (Uhe et al, 2016). Please add the equivalent of Figs 1-3 for the variance, preferably of daily data but monthly should be OK if these were not saved. In that case CRU-TS can also be used as ground truth, for daily data Berkeley Earth has temperature fields and CPC precipitation fields over the required period.

3. Section 4.3. It would be useful to explicitly comment to what extend the biases in extremes can be corrected by a simple additive (temperature) or multiplicative (precipitation) bias correction.

4. Section 4.4 Given the strong connection between the reliability and trends, pleae sadd trend maps of the observations and model results in addition to the reliability diagrams, preferably also with SLP trends.

Minor comments

p.5 l.30 Why is Z500 taken from the ancient ERA40 reanalysis rather than a more modern one? JRA-55 covers the period 1961-1990.

p.6 l.8 "30 years period from 1961–1990". I understand that this is dictated by the short runs of w@h1. Can you add a comment on how different the biases of w@h2 are over the whole century?

Almost all figures would be more intuitive for readers with a left-to-right script if w@h1 was plotted to the left of w@h2.

Please show Fig. S1 in the main text instead of Fig.3 as it is much more informative.

p.7 l.8 "suggesting that certain modes are not well represented". To be nit-picking: misrepresentation of modes will affect the variability much more than the mean state. Just delete, as it carries no useful information.

p.7 l.22-31. You should mention that by prescribing SST you pretty much fix the trends over land as well (eg Shin et al, Clim.Dyn. 2011 and other papers from Sardeshmukh's

group). The agreement is therefore not all that surprising.

p.7 l.22-31 Some formal analysis how many times the temperature falls outside the ensemble range seems called for, ie whether the ensemble is reliable: is the spread a good representation of variability? Note that this is not covered in section 4.4, as there the distributions are normalised to their own variability.

p.7 l.32- The same holds for the regional time series.

p.11 l.2 "and may be the subject of further work" is not useful information.

p.11 l.14 Why did you not take a standard percentile for the shading, like the 95% CI, rather than the full range of 1000 bootstrap sample?

p.11 l.29 I am also not impressed by the cold extremes in France and the British Isles, especially with the non-linear behaviour there.

p.12 l.19 Can you make the connection between the "attribution of extreme weather events" and "seasonal temperature in the upper tercile" more explicit? What are the reasons to assume that if the model is reliable in the latter it is suitable for the former?

p.12 l.20 It is not clear to me whether these reliability diagrams are computed using all grid points in the region, as the Met Office group does, or using the area-averaged value for the region. Please clarify.

Fig.13 Please explain the difference between the red and green dots.

p.12 l.31 How does this assessment that the model performs well after calibration compare to publications that w@h1 and other RCMs are very poor at simulating trends in heat waves (Min et al, 2013; Sippel et al, 2016)?

p.13 l.6 "For low summer precipitation (Fig. 15), the reliability is found to be rather good in IP, AL, EA, ME" I do not see that by eye. Please use a more objective criterion, such as the fit by Weisheimer and Palmer (2014).

p.13 l.15 "Therefore, these results may be dominated by the long-term trend arising from increased greenhouse gas concentrations," This is fairly certain, as seasonal predictability in Europe is dominated by the trend.

p.14 l.30 "Overall, weather@home is an excellent tool for the investigation of extreme weather events." should read "may be a useful tool if proper bias corrections and other caveats are taken into account". As with every climate model.
* * *

---

## Author Comment (AC1) · 3 Mar 2017

**Reply to the review by Anonymous Referee #1**

We are very grateful to the anonymous referee #1 for the constructive and positive review. We include our answers to the comments in blue font right under the unmodified comments from the review. We note that the line numbers provided by reviewer #1 refer to the originally submitted document and do not correspond to the published discussion manuscripts. When referring to a particular line in our answer here, we provide line numbers for both documents to avoid confusion.

**REVIEW**

The manuscript describes the validation of the improved global-regional climate modeling system weather@home2. Reading this manuscript has been a real pleasure! The manuscript describes in a concise and very well written way the changes compared to the previos version of the model system and their impact on the global and regional climate over Europe. Figures illustrate the important results. I can recommend this manuscript for publication after the authors have addressed a few questions and comments.

We thank the referee for his positive and encouraging review.

Abstract and Conclusions: I wouldn't fully agree to the statement that European biases are reduced. It is certainly true for the temperature, but precipitation? Look at Fig.9, the w@h1 0.22 deg results are often better than w@h2! I suggest you differentiate between temperature and precipitation biases.

We agree that for precipitation the improvement in w@h2 is not clear, however we think the different resolution of the two models leads to some confusion: the w@h1 0.22 degree and the w@h2 0.44 degrees results are based on interpolated/aggregated data, as the models are not run at these resolutions. From Figure 9, root-mean square biases are reduced when precipitation is either interpolated or aggregated, which highlights an issue of precipitation location in both models. Hence, the apparent better performance of w@h1, 0.22 degree may be just an artefact. This was explained in section 4.1, from line 21 on page 9 until the end of section 4.1.

Table 2 in the paper lists the mean regional bias for both models in the sub-regions. w@h2 is better than w@h1 in many cases as well, but indeed there is overall not a clear improvement compared to w@h1. Therefore, we follow the referee's suggestion and change the following sentence in the abstract:

"The European RCM biases are overall reduced, in particular the warm and dry bias over eastern Europe, but large biases remain"

to

"The European RCM temperature biases are overall reduced, in particular the warm

bias over eastern Europe, but large biases remain. Precipitation is improved over the Alps in summer, with mixed changes in other regions and seasons."

And we have added the following sentence in the conclusion: "Precipitation biases in HadRM3P, on the other hand, do not exhibit substantial improvements overall".

Sec2.3: it is not clear to me how you create the initial conditions for each simulation. Are single-year spinup simulations part of the 13-mnth long experiment (i.e. making it 25 month long), or how exactly is it done? Please explain.

We agree with the referee that this needs clarification. Two separate sets of simulations were run. A first set of 12-months long simulations (December to November) have been run in a first experiment in order to create spun-up conditions (the initial state for the spin-up simulations comes from a long HadAM3P simulation with MOSES 1 and was reconfigured to MOSES 2). The end state from the spin-up simulations were then used to initialise the 13 months long experiment. We have clarified this in the first paragraph of Section 2.3, which now reads:

"A large ensemble of w@h2 consisting of more than 100 simulations per year from 1900–2006 is analysed. First, a restart file from a century-long HadAM3P simulation with MOSES 1 has been reconfigured for MOSES 2. This initial condition file is then used in a spin-up ensemble consisting of 12-month simulations (from December to November, with multiple simulations for each year), providing spun-up initial conditions on December 1st each year. The simulations analysed in this paper are then initialised on the 1st of December each year from the end state of the spin-up ensemble and are run for 13 months. (...)".

Sec2.3: How good is the initialization of soil and vegetation variables? Soil has a memory in excess of 1 year, so a 1-yr spin-up may not be sufficient for soil temperature and humidity. You have made a large effort to improve the land surface and vegetation components in your model, yet an inaccurate initialization could make these improvements worthless. Could you comment on that?

This is a very good point. The initial conditions used for the spin-up simulations are derived from a multi-decadal HadAM3P simulation. The land surface model in that simulation was MOSES 1, therefore the soil initial conditions are spun-up to that model. As the referee correctly points out, one year is however a rather short spin-up to the more recent land surface model MOSES 2, although one might expect these to be not too different. Unfortunately soil temperature was not saved as an output in these simulations, but we have looked at soil moisture and could find a small spin-up effect from our simulation output.

Fig. R1 and R2 display the difference in soil moisture between end of year 2 and end of year 1 (monthly average in December, which are months 13 and 25 from the restart with MOSES 1 conditions) for the 4 soil layers, scaled by the standard deviation of soil moisture at the end of year 2 (i.e., month 25). In some regions, large changes are

found (GCM: North Africa in all layers, and Asia/Western North America in the deepest layer, RCM: mostly only the deepest layer in Europe). This suggests that the soil has partly, but perhaps not fully equilibrated with the model. Fortunately, the upper 1m of the soil, corresponding to the root zone in most regions and therefore most critical for evapotranspiration, appears relatively well spun-up over Europe. Unfortunately, it is not possible to assess whether an additional year would lead to further changes, as these are not available.

To nonetheless test the spin-up effect on our analysis, we display the biases in temperature and precipitation in HadAM3P and HadRM3P for both years in Figs. R3–R6. The largest impact is found in DJF but is unlikely due to soil moisture as it spans all latitudes. For temperature, the most striking difference is an reduction of the bias over Southeast Europe, which may be driven by increased soil moisture in this region and possibly by effects of soil temperature. This suggests that a longer spin-up might potentially further reduce this model bias and thus that the spin-up may not be sufficient. For precipitation, the impact is small globally, in all seasons except DJF and, in other seasons, in Sahara, where % biases are very sensitive to small changes. DJF impacts are found throughout latitudes and are thus unlikely to be a soil moisture spin-up issue but may results from changes in circulation induced by temperature changes. In HadRM3P, similar results are found, with mostly an impact in DJF unlikely related to soil moisture.

These results highlight that a longer spin-up may be required in future uses of w@h2. In light of these results, we plan to update the w@h2 experimental setup to use spun-up conditions from longer simulations.

We have therefore included these figures in the Supplementary Information (Supplementary Figs. S5–S7 and S15–S17), and have added the following comments in the main text of the paper:

- Section 2.3: "The effect of the relatively short spin-up for soil variables on simulated temperature and precipitation is discussed in Sect. 3.1 for HadAM3P and Sect. 4.1 for HadRM3P".
- Section 3.1: "Finally, to assess whether the 1-year spin-up was sufficient to allow the soil variables to be spun-up, Supplementary Fig. S5 shows the difference between ensemble mean soil moisture (for each soil layer) in December between the 1st month and the 13th month of the analysed simulations (i.e., 13th and 25th month of simulation, respectively), scaled by the standard deviation of the second one. Apart from North Africa, the differences are confined to the 3rd (Central Asia) and 4th layer (many regions). This suggests that a longer spin-up may required in future experiments with w@h2. Fortunately, however, the upper 1m of the soil, corresponding to the root zone in most regions and therefore most critical for evapotranspiration, appears relatively well spun-up over Europe. It is not possible to assess whether an additional year would lead to further changes, as these are not available, and soil temperature is not examined here as this variable has not been saved in our simulations. The impact on temperature biases is shown in Supplementary Fig. S6 and the largest impact is found in DJF but is unlikely due to soil moisture as it spans all latitudes. The most striking difference is a reduction of the bias over Southeast Europe and Central US, which may be driven by increased soil moisture in these regions with soil moisture-limited evapotranspiration regimes (Seneviratne et al., 2010) and possibly by effects of soil temperature. An impact is also found in MAM. This suggests that a longer spin-up might potentially further reduce the summer temperature warm model bias. For precipitation (Supplementary Fig. S7), the impact is small globally, in all seasons except DJF and, in other seasons, over Sahara (note that % biases are very sensitive to small changes in this region). DJF impacts are found throughout latitudes and are thus unlikely to be a soil moisture spin-up issue but may result from changes in circulation induced by temperature changes. These results highlight that a longer spin-up may be required in future uses of w@h2, which will be implemented for future w@h2 experiments."

- Section 4.1: "Finally, the impact of the short spin-up is evaluated as was done in Sect. 3.1 for HadAM3P. Fig. S15 shows the difference in soil moisture as in Fig. S5 (see Sect. 3.1). Over Europe, only Finland and Northwestern Russia display large differences in the upper 1 m of the soil. In the deepest layer, soil moisture is larger in the analyzed year than in the previous year over Southeastern Europe and in some other regions, but this deep layer is less critical to evapotranspiration and therefore to surface climate. Analysis of temperature and precipitation biases (Figs. S16 and S17) show that the hot MAM and JJA biases over Southeastern Europe are reduced with progressing spin-up, as expected from the increasing soil moisture and suggesting that a longer spin-up may further reduce this bias. Temperature biases in DJF and precipitation biases in all seasons are not related to soil moisture changes in a straightforward manner, and hence could be due to soil temperature, a variable not saved as an output in our simulations and therefore not analysed here.".
- Conclusion (Section 5): "A limitation of w@h2 as presented in this study is the relatively short spin-up (1 year). We find that a longer spin-up may further improve w@h2, in particular with respect to the representation of summer temperatures over Southeastern Europe. Future w@h2 experiments will therefore include a longer spin-up of 5–10 years, in order to allow for a full stabilization of soil moisture and soil temperature and to thereby take full advantage of the capability of the model."

Sec4.4: To be honest, I was somewhat surprised to see a section about reliability in this manuscript. Reliability is a very specific term with a precise definition in the verification of probabilistic forecasts, but I have never encountered it in the context

of climate simulations. On the other hand, the reliability of climate models is often discussed (e.g. in the IPCC AR) in the casual meaning of reliability as a synonym to trustworthiness. In this second definition of reliability, one often looks at how well the pdf of a quantity from a climate model matches the observed distribution. I wonder if this latter approach was what you had in mind when you started discussing reliability. Reliability and attribution diagrams as you present them now don't make much sense in the context of climate simulations, they should only be used for the verification of probabilistic forecasts. I therefore suggest you remove section 4.4 completely.

While we agree that it is slightly unusual to use reliability diagrams in this context, we do not agree that they don't make sense. We believe that Section 4.4 provides a useful quantification of the ability of weather@home2 to realistically simulate the response of climate to its drivers, which is very relevant for attribution. Given this and since Referee #2 showed interest in the reliability results, we have decided to keep this section. In addressing the comments from referee #2, we have complemented it with an new figure showing regional trends to support the interpretation of reliability diagrams, and have added some quantitative statements on model's reliability based on Weisheimer and Palmer (2014).

Figure R1: Soil moisture spin-up in HadAM3P. Difference between ensemble mean soil moisture in December between the end of the 1st year ("spin-up", 13th month from the generic restart) and the end of the 2nd year (25th months from the generic restart) in each simulation, normalized by the standard deviation (taken from the end of the 2nd year). Years 1961–1990 were used.

**References**

- Seneviratne, S. I., Corti, T., Davin, E. L., Hirschi, M., Jaeger, E. B., Lehner, I., Orlowsky, B., and Teuling, A. J.: Investigating soil moisture-climate interactions in a changing climate: A review, Earth-Sci. Rev., 99, 125–161, doi:10.1016/j.earscirev.2010.02.004, URL http://linkinghub.elsevier.com/retrieve/pii/S0012825210000139, 2010.
- Weisheimer, A. and Palmer, T. N.: On the reliability of seasonal climate forecasts, J. R. Soc. Interface, 11, doi:10.1098/rsif.2013.1162, URL http://rsif.royalsocietypublishing.org/content/11/96/20131162, 2014.

---

## Author Comment (AC2) · 3 Mar 2017

**Reply to the review by Anonymous Referee #2**

We thank anonymous referee #2 for the positive review, which suggests useful additions to the manuscripts. These suggestions are highly appreciated. Answers to the comments are included in blue font right under the unmodified comments from the review.

**REVIEW**

Review of 'weather@home 2: validation of an improved global-regional climate modelling system' by Guillod et al. The paper is a useful and fairly thorough documentation of the w@h system with a focus on Europe, although some gaps remain that can easily be filled. Also, there are some qualitative statements that can be converted to quantitative ones with, I think, relatively minor effort. I am looking forward to a more complete version, which would be very informative.

We appreciate the overall positive tone of the reviewer's comments as well as the relevant points raised, for which we mention our intentions for the revised manuscript.

**Major comments**

1. Can you comment how the biases in the global model compare to other, state-of-the-art, GCMs, eg in Chapter 9 of the IPCC WG1 AR5?
   Following the referee's comment, we have had a closer look at w@h biases and reproduced a few figures of the mentioned IPCC chapter for our model. Figs. R1, R2 show the absolute error in annual-mean temperature and precipitation, and can be directly compared (visually) to the following figures in Chapter 9 of the IPCC WG1 AR5: 9.2(c) and 9.4(c), respectively. Overall and for both variables, HadAM3P performs similarly well to state-of-the-art CMIP5 models (despite being an older model, but possibly because of the prescribed SSTs). Larger errors in temperature are however found in HadAM3P over Greenland and Eastern North America. For precipitation, no such hotspot or error is found. For a regional and seasonal quantification, we have reproduced Fig. 9.39 for SREX regions (Fig. R4, but note that the regions in a different order than the IPCC AR5 Fig. 9.39). Here, too, HadAM3P performance looks similar to the CMIP5 models, with a few cases where HadAM3P biases are larger. We have therefore added the following two sentences, at the end of the two paragraphs dealing with temperature and precipitation biases in Section 3.1: "For most regions, the performance of HadAM3P is similar to state-of-the-art coupled climate models from CMIP5 (Flato et al., 2013)" and "Like with temperature, the model performs similarly to typical CMIP5 models (Flato et al., 2013)".

2. For attribution, a correct representation of variance is as important as trends (Uhe et al, 2016). Please add the equivalent of Figs 1-3 for the variance, preferably of daily data but monthly should be OK if these were not saved. In that case CRU-TS can also be used as ground truth, for daily data Berkeley Earth has temperature fields and CPC precipitation fields over the required period.

   Daily data were not saved in the global model. Therefore, we have followed the referee's suggestion and have computed bias maps of the standard deviation of monthly averaged temperature, precipitation and 500hPa geopotential height. Due to the large number of figures in the paper, these are placed in the Supplementary Information (Figs. S1, S3 and S4), but the main text (Section 3.1) mentions the main results from these.

3. Section 4.3. It would be useful to explicitly comment to what extend the biases in extremes can be corrected by a simple additive (temperature) or multiplicative (precipitation) bias correction.

   We thank the referee for this useful suggestion. We have added some comments on bias correction and the suitability of various techniques in the last paragraph of Section 4.3, with a direct reference to the quantile-quantile plots.

4. Section 4.4 Given the strong connection between the reliability and trends, please add trend maps of the observations and model results in addition to the reliability diagrams, preferably also with SLP trends.

   We appreciate the referee's suggestion to add information about trends. However, we have found that trend maps are highly variable within the ensemble, and using the ensemble mean leads to a spuriously smoothed spatial pattern, as the effect of internal variability is removed. Therefore, rather than trend maps, we have chosen to show trends for regional averages of temperature and precipitation (Fig. 17), which allows us to display the spread in trends from individual w@h2 time series (constructed by randomly sampling 1 ensemble member per year). We have also added some text in Section 4.4 related to that figure, and have renamed that section "Reliability and trends".

**Minor comments**

- p.5 l.30 Why is Z500 taken from the ancient ERA40 reanalysis rather than a more modern one? JRA-55 covers the period 1961–1990.
  We have replaced ERA40 with JRA-55 in the manuscript.

- p.6 l.8 "30 years period from 1961–1990". I understand that this is dictated by the short runs of w@h1. Can you add a comment on how different the biases of w@h2 are over the whole century?
  We have also plotted the w@h2 bias maps for the time period century (1900–2006), and they look very similar to the ones with years 1961–1990. We have

therefore added the following sentence at the beginning of Sect. 3: "w@h2 biases look very similar when the whole time period, from 1900–2006, is considered".

- Almost all figures would be more intuitive for readers with a left-to-right script if w@h1 was plotted to the left of w@h2.
  We follow the referee's suggestion in all figures.

- Please show Fig. S1 in the main text instead of Fig.3 as it is much more informative.
  We agree that precipitation bias maps tend to over-represent wet regions when shown in mm/day. However, relative biases (in %) tend to over-represent dry regions in a similar way. After having tested both ways, we would like to keep this as is since we find that this is best for the discussion of the biases in Section 3.1.

- p.7 l.8 "suggesting that certain modes are not well represented". To be nit-picking: misrepresentation of modes will affect the variability much more than the mean state. Just delete, as it carries no useful information.
  The sentence was replaced with "The bias patterns are similar in both models w@h1 and w@h2".

- p.7 l.22-31. You should mention that by prescribing SST you pretty much fix the trends over land as well (eg Shin et al, Clim.Dyn. 2011 and other papers from Sardeshmukh's group). The agreement is therefore not all that surprising.
  We have added the following sentence, albeit with a cautious formulation due to our new findings of Section 4.4, which show that local to regional trends exhibit large variability depending on the ensemble members and hence do not appear that strongly constrained by SSTs: "Although this may not be surprising since others have found that prescribing SSTs may strongly force trends over land (e.g., Shin and Sardeshmukh, 2011), we note that regional trends computed from various ensemble members suggest a large range of trends despite the prescription of SSTs (see Sect. 4.4)."

- p.7 l.22-31 Some formal analysis how many times the temperature falls outside the ensemble range seems called for, ie whether the ensemble is reliable: is the spread a good representation of variability? Note that this is not covered in section 4.4, as there the distributions are normalised to their own variability.
  Thank you for this good suggestion. We have added to the anomaly time series (Figs. 5, S9 and S11) the fraction of years when the observation lies within the 5–95% confidence interval of the w@h2 ensemble. For the global time series these are 71% for temperature and 58% for precipitation. We have added the following sentences in Section 3.2: "For temperature ... CRU-TS mostly lies within the 90% confidence interval of the w@h2 ensemble (71% of the years, suggesting that variability at the global scale might be slightly underestimated)...

For precipitation... CRU-TS appears to lie more often outside the w@h2 ensemble for precipitation than for temperature (observed values are within the 5–95% range from w@h2 on only 58% of the years)...".

- p.7 l.32- The same holds for the regional time series.

  As mentioned above, this has also been added to the regional time series figures (Supplementary Figures S9 and S11).

- p.11 l.2 "and may be the subject of further work" is not useful information.

  Removed.

- p.11 l.14 Why did you not take a standard percentile for the shading, like the 95% CI, rather than the full range of 1000 bootstrap sample?

  We thank the referee for his suggestion. We have changed our plots to show the 95% confidence interval from the bootstrap samples. Besides the description of this in Sec. 4.3 and in the respective figure captions, no text was changed as the results are qualitatively the same.

- p.11 l.29 I am also not impressed by the cold extremes in France and the British Isles, especially with the non-linear behaviour there.

  Yes indeed. We have added the following sentence: "Extreme cold night in BI and FR, however, are also underestimated by the model (i.e., extreme cold night are not cold enough)".

- p.12 l.19 Can you make the connection between the "attribution of extreme weather events" and "seasonal temperature in the upper tercile" more explicit? What are the reasons to assume that if the model is reliable in the latter it is suitable for the former?

  Our reliability analysis focuses on seasonal averages, not on extreme weather events as such. However, both are related to some extent, as extreme weather events can have a significant impact on the seasonal average. In addition, if a specific set of forcings (greenhouse gases, SST pattern, ...) is conducive to higher temperature, it will lead to higher seasonal averages and likely also hotter heat waves. We have added the following sentence in the first paragraph of Section 4.4 to emphasize these points: "While seasonal averages are not directly related to extreme weather events, the drivers of both are likely similar (e.g., higher $CO_2$ leads to increased mean and extreme temperature), and the occurrence of a few extreme events may strongly impact the seasonal average".

- p.12 l.20 It is not clear to me whether these reliability diagrams are computed using all grid points in the region, as the Met Office group does, or using the area-averaged value for the region. Please clarify.

  The reliability diagrams use area-averaged values for the region. This was clarified by adding "regional area-averaged" in the following sentence of section 4.4:

"For each type of event (e.g., high summer temperature, defined as JJA averaged temperature in the upper tercile), the probability of the event is computed for each year from **regionally averaged** w@h2 model output ("forecast probability")".

- Fig.13 Please explain the difference between the red and green dots.
  There is no green dot on Figs. 13–16, so we assume this refers to the black dots. As explained in the figure caption, "bins containing less than five years shown in black". Red dots, on the other hand, are for bins with at least five years. This was clarified in the caption by inserting "(red dots indicate bins containing at least 5 years)". It should be mentioned that black dots were, therefore, mostly not considered in the description of the results as they do not correspond to robust values. We have added a sentence to make the reader aware of this in the main text: "Results for bins containing at least 5 data points (i.e., years) are shown in red, while for other bins, shown in black, values are not very robust and should be interpreted with caution".

- p.12 l.31 How does this assessment that the model performs well after calibration compare to publications that w@h1 and other RCMs are very poor at simulating trends in heat waves (Min et al, 2013; Sippel et al, 2016)?
  The reliability analysis is based on seasonal averages, while heat waves usually last a few days. We have not investigated trends in heat waves specifically in our analysis. Nonetheless, the two studies mentioned by the referee (Min et al., 2013; Sippel et al., 2016) point to an underestimation in heat wave trends by RCMs compared to observations. This is nicely consistent with the "underconfidence" that we find for hot summer: the model's sensitivity to greenhouse gas forcings may be too low. We therefore added the following sentence: "Interestingly, this underestimation of the sensitivity of hot temperatures to forcings is consistent with the tendency of RCM to underestimate trends in heat waves over Europe (Min et al., 2013; Sippel et al., 2016)"

- p.13 l.6 "For low summer precipitation (Fig. 15), the reliability is found to be rather good in IP, AL, EA, ME" I do not see that by eye. Please use a more objective criterion, such as the fit by Weisheimer and Palmer (2014).
  We thank the referee for this useful suggestion. We have implemented the fit and bootstrap sampling proposed by Weisheimer and Palmer (2014) and we now display their proposed categorisation on the upper left of each plot in Figures 13–16 (note that as more simulations have been completed since our initial submission, some of the figure have changed slightly). We have also added a table (Table 3) which summarises the five categories. For temperature, very good performance is found, with categories 4 and 5 in almost all cases. For precipitation, performance is much lower, with categories 1–3 being most prominent. We have substantially edited Section 4.4 to include the information provided by this met-

**yearly mean absolute tas bias**

[Figure]

Figure R1: HadAM3P bias in yearly mean 2m-temperature with respect to CRU-TS (degrees C).

ric.

- p.13 l.15 "Therefore, these results may be dominated by the long-term trend arising from increased greenhouse gas concentrations", This is fairly certain, as seasonal predictability in Europe is dominated by the trend.
  We have changed "may be" to "are".

- p.14 l.30 "Overall, weather@home is an excellent tool for the investigation of extreme weather events." should read "may be a useful tool if proper bias corrections and other caveats are taken into account". As with every climate model.
  Done.

**References**

Flato, G., Marotzke, J., Abiodun, B., Braconnot, P., Chou, S. C., Collins, W. J., Cox, P., Driouech, F., Emori, S., Eyring, V., et al.: Evaluation of Climate Models. In: Climate Change 2013: The Physical Science Basis. Contribution of Working Group I to the Fifth Assessment Report of the Intergovernmental Panel on Climate Change, Climate Change 2013, 5, 741–866, 2013.

Min, E., Hazeleger, W., van Oldenborgh, G. J., and Sterl, A.: Evaluation of trends in high temperature extremes in north-western Europe in regional climate models, Environ Res Let, 8, 014 011, URL http://stacks.iop.org/1748-9326/8/i=1/a=014011, 2013.

**yearly mean absolute pr bias**

[Figure]

Figure R2: HadAM3P bias in yearly mean precipitation with respect to CRU-TS (mm/day).

Shin, S.-I. and Sardeshmukh, P. D.: Critical influence of the pattern of Tropical Ocean warming on remote climate trends, Clim Dyn, 36, 1577–1591, doi:10.1007/s00382-009-0732-3, URL `http://dx.doi.org/10.1007/s00382-009-0732-3`, 2011.

Sippel, S., Otto, F. E. L., Flach, M., and van Oldenborgh, G. J.: The role of anthropogenic warming in 2015 Central European heat waves [in Explaining extreme events of 2015 from a climate perspective], Bull Am Meteorol Soc, 97, 551–556, 2016.

Weisheimer, A. and Palmer, T. N.: On the reliability of seasonal climate forecasts, J. R. Soc. Interface, 11, doi:10.1098/rsif.2013.1162, URL `http://rsif.royalsocietypublishing.org/content/11/96/20131162`, 2014.

[Figure]

Figure R3: HadAM3P relative bias in yearly mean precipitation with respect to CRU-TS (%).

[Figure]

Figure R4: HadAM3P biases in (left) temperature and (right) precipitation in DJF (top), JJA (middle) and annually (bottom), for the SREX regions. As Fig. 9.39 of Chapter 9 of IPCC WG1 AR5.

---

## Author Response (AR2)

**Reply to the Topical Editor**

Dear Editor,
We thank you very much for your review of our revised manuscript. We include our answers to the comments in blue font right under the unmodified comments from the Editor, followed by the revised manuscript with tracked changes.

**EDITOR REVIEW**

Dear authors,

I read the paper and your responses. I appreciate the detail to which you responded to the spinup concerns of reviewer 1 and the quantification of trends and bias in the standard deviations by reveiwer 2. I will accept the paper, including the supplement, pending some minor revisions.

We thank the Editor for recognizing our efforts and for the additional minor comments, which we address below.

These include the following:

1) Although layer 1 of the soil seems spun up after a year, the lower layers are not. You describe that adequately, but the statement that only the root zone is relevant is a bit at odds with the need for the model improvements. For longer term the spinup in the deeper soil could have an effect, but we cannot know with these experiments. Could you rephrase a little bit?

We agree mostly with the Editor, since spinup of the deeper soil could affect the dynamics in upper layer, however we stand by our statement that the root zone is most critical to land-atmosphere fluxes. We have therefore added some text as follows:

- Page 8 lines 3–4: "Nonetheless, the soil moisture state in deeper layers may in some cases impact soil moisture dynamics in the root zone and, thereby, affect land-atmosphere exchange and surface climate".

- Page 11, line 14: change "but this deep layer is less critical to evapotranspiration" to "but this deep layer may be less critical to evapotranspiration".

2) The comparison to CMIP5 coupled models is not fair. SST is prescribed here, so the model would always do a reasonable job in many meteorological parameters (at least averages). So, the comment should be made more careful and it would be good to have a reference to CMIP5 AMIP results here instead of the coupled results.

The editor is of course right: the comparison with CMIP5 is not entirely fair. However, since the IPCC AR5 Chapter 9 on model validation (Flato et al., 2013) does not

display biases from AMIP simulations (coupled simulations only), we cannot repeat the direct, simple comparison of our model biases that we had done in response to the comment from referee #2 in the previous round of reviews using AMIP simulations. Therefore, we have kept our statements, but have added clear mention of this comparability issue in the relevant sentences as follows:

- Page 6, line 30: after "For most regions, the performance of HadAM3P is similar to state-of-the-art coupled climate models from CMIP5 (Flato et al., 2013)", we have added ", although a fair comparison is difficult given that in w@h2 the ocean state is prescribed to observations while in CMIP5 models it is computed interactively by an ocean model coupled to the atmospheric model".

- Page 7, line 12-13: we added "but note that as for temperature, this comparison may not be fair given the prescribed SSTs in w@h2 as opposed to interactive ocean in CMIP5".

3) GMD supports open access of code and data. Your code is not available and the link to the workshop does not work. A persistent identifier to the code would be preferable, even if the model software liscence doesnt allow the software to be open (e.g. via ZENODO) and a link that does not break as well. Moreover, is it possible to make the data on which this model evaluation is based open? For instances via DATACITE you can make your data findable.

Thank you for highlighting these issues of open data and softwares, which we fully agree are important. The link to the workshop was changed recently and it has now been updated in the manuscript – many thanks for noticing that the link did not work. Unfortunately, the Met Office license does not allow us to share the model code. On the other hand, we can guarantee that the data used in this study (output of the model simulations) will be made available on the CEDA archive (`http://www.ceda.ac.uk`), following an agreement between the climateprediction.net team and CEDA. Since the process of making data available on CEDA is taking longer than expected, we now state in the manuscript that data will be made available in the near future and that access to the data can be asked by email in the meantime (page 17, line 20). When published on the portal, the dataset will refer to this paper on the CEDA archive.

**References**

[revised manuscript text omitted]